# Microvascular stabilization via blood-brain barrier regulation prevents seizure activity

Chris Greene [1], Nicole Hanley [1], Cristina R. Reschke [2,3], Avril Reddy[1], Maarja A. Mäe [4], Ruairi Connolly[2,5,6], Claire Behan[2,5,6], Eoin O'Keeffe[1], Isobel Bolger[1], Natalie Hudson[1], Conor Delaney [1], Michael A. Farrell[7], Donncha F. O'Brien[8], Jane Cryan[7], Francesca M. Brett[7], Alan Beausang[7], Christer Betsholtz[4], David C. Henshall[2,9], Colin P. Doherty[2,5,6] & Matthew Campbell [1,2✉]

Blood-brain barrier (BBB) dysfunction is associated with worse epilepsy outcomes however the underlying molecular mechanisms of BBB dysfunction remain to be elucidated. Tight junction proteins are important regulators of BBB integrity and in particular, the tight junction protein claudin-5 is the most enriched in brain endothelial cells and regulates size-selectivity at the BBB. Additionally, disruption of claudin-5 expression has been implicated in numerous disorders including schizophrenia, depression and traumatic brain injury, yet its role in epilepsy has not been fully deciphered. Here we report that claudin-5 protein levels are significantly diminished in surgically resected brain tissue from patients with treatment-resistant epilepsy. Concomitantly, dynamic contrast-enhanced MRI in these patients showed widespread BBB disruption. We show that targeted disruption of claudin-5 in the hippocampus or genetic heterozygosity of claudin-5 in mice exacerbates kainic acid-induced seizures and BBB disruption. Additionally, inducible knockdown of claudin-5 in mice leads to spontaneous recurrent seizures, severe neuroinflammation, and mortality. Finally, we identify that RepSox, a regulator of claudin-5 expression, can prevent seizure activity in experimental epilepsy. Altogether, we propose that BBB stabilizing drugs could represent a new generation of agents to prevent seizure activity in epilepsy patients.

[1] Smurfit Institute of Genetics, Trinity College Dublin, Dublin 2, Ireland. [2] FutureNeuro, Science Foundation Ireland Research Centre for Chronic and Rare Neurological Diseases, Royal College of Surgeons in Ireland, University of Medicine and Health Sciences, Dublin, Ireland. [3] School of Pharmacy and Biomolecular Sciences, RCSI University of Medicine and Health Sciences, Dublin 2, Ireland. [4] Department of Immunology, Genetics, and Pathology, Rudbeck Laboratory, Uppsala University, Uppsala, Sweden. [5] Department of Neurology, Health Care Centre, Hospital 5, St James's Hospital, Dublin 8, Ireland. [6] Academic Unit of Neurology, Room 5.41, Biomedical Sciences Institute, Trinity College Dublin, Dublin 2, Ireland. [7] Department of Neuropathology, Beaumont Hospital, Dublin, Ireland. [8] Department of Neurosurgery, Beaumont Hospital, Dublin, Ireland. [9] Department of Physiology and Medical Physics, RCSI University of Medicine and Health Sciences, Dublin 2, Ireland. ✉email: matthew.campbell@tcd.ie

Epilepsy affects up to 70 million people worldwide and is characterised by recurrent unprovoked seizures[1]. While there are effective medicines to treat seizures, up to one third are refractory to anti-convulsant drugs[2]. There is an urgent need therefore, to identify ways to convert otherwise drug-resistant individuals to drug-responsive. While some conditions such as Dravet syndrome represent a well-defined genetic cause of epilepsy, in the majority of cases, the cause of epilepsy is unknown[3]. We do however know that conditions such as stroke, brain tumours, and traumatic brain injury can lead to the development of symptomatic seizure activity and lead to epilepsy[4,5]. Added to this, it is now well established that disruption to the microvasculature of the brain is a common feature in each condition and specifically, a breakdown in the integrity of the so-called blood-brain barrier (BBB) appears to be a key driver of pathology. Seizures and spontaneous ictal events, typical of epilepsy, have been observed in experimental animal models following focal BBB insult and the degree of BBB disruption strongly correlates with seizure severity[6–9]. The BBB is formed by endothelial cells lining the cerebral microvasculature. Its principal components are inter-endothelial tight junction proteins which seal the paracellular space and both luminal and abluminal proteins controlling trans-endothelial movement of carbohydrates, hormones, ions and metabolic by-products[10]. Previous studies in animal models of epilepsy and in resected brain tissue from pharmacoresistant patients has revealed a plethora of abnormalities of the BBB, including tight junction disruption, increased immune cell infiltration and aberrant angiogenesis[11–17]. While ample studies exist describing BBB dysfunction in animal models of epilepsy, there is little understanding of the underlying molecular drivers of BBB disruption, which means that targeted regulation of BBB dysfunction is a tantalising but unexplored therapeutic channel for novel epilepsy treatment.

Claudin proteins are key structural components of tight junctions. Several claudins (−1, −3, −5 and −12) were originally thought to be expressed in the BBB but more recent studies indicate claudin-5 is the dominant component with limited expression and contribution of other claudins in maintaining homoeostasis and BBB integrity[18–20]. Indeed, it has been estimated that a single brain endothelial cell will express up to 18 million claudin-5 molecules (Mäe et al., manuscript in preparation). Commensurate with a critical role in brain function, claudin-5 knockout mice die within 10 h of birth[21]. We have previously described the generation and phenotyping of an inducible claudin-5 knockdown mouse[22]. These animals display a polypathology, showing evidence of cognitive decline, loss of acoustic pre-pulse inhibition and increased anxiety. They also phenocopy the knockout mouse in that prolonged claudin-5 depletion leads to death. However, in the days prior to death, the mice show evidence of seizure activity, suggesting that claudin-5 levels are critical to seizure development.

Here, we show for the first time, a direct correlation between human epilepsy and claudin-5 levels at the BBB. Using a range of advanced imaging paradigms in human subjects coupled to 4 distinct animal models, we show that focal and brain-wide depletion of claudin-5 results in seizures in mice. We further link tight junction dysfunction to aberrant immune cell interaction at the BBB and the development of chronic neuroinflammation. Importantly, we show that restoration of claudin-5 levels attenuates seizures and neuroinflammation. Finally, we show that via small molecule inhibition of ALK5 and the TGF-beta receptor, BBB integrity can be stabilised as a result of ALK5 inhibition and claudin-5 up-regulation. As tight junctions regulate the local ionic environment, we hypothesise that tight junction stabilisation, via claudin-5 targeting, may represent a unique way of controlling seizure activity, especially in treatment resistant patients.

## Results

**Blood-brain barrier dysfunction in the temporal lobe of treatment-resistant epilepsy patients.** We collected sclerotic resected brain tissue from 16 patients with treatment resistant temporal lobe epilepsy (TLE) of which 4 patients underwent dynamic contrast-enhanced magnetic resonance imaging (DCE-MRI) prior to their surgery (Supplementary Table 1 for demographics). 3T MRI revealed mesial temporal sclerosis in 2/4 patients and left temporal atrophy in the other 2 patients. Subsequently, the patients underwent an anterior temporal lobectomy (3/4) or selective amygdalohippocampectomy (1/4) with postsurgical neuropathological examination revealing sub-pial gliosis (2/4), pyramidal neuron cell loss (1/4) and neuronal depletion, dispersion and CA1 cell loss (1/4) (Supplementary Table 1). DCE-MRI imaging revealed numerous sites of gadolinium extravasation in individuals with TLE with and without hippocampal sclerosis (Fig. 1a–c). In contrast, age-matched controls showed few signs of gadolinium extravasation in the temporal lobe or surrounding regions (Supplementary Fig. 1a). There was extensive neuroinflammation in the hippocampus of TLE brains with increased expression of ICAM1, CCL2, GFAP, TNF and IL1B mRNA compared to autopsy control tissue and unaffected cortex tissue (Supplementary Fig. 1c). Sections of resected tissue showed a decrease in the pattern and intensity of claudin-5 staining relative to the endothelial cell marker cluster of differentiation 31 (CD31) (Fig. 1d, f; Supplementary Table 2 for demographics), with immunoglobulin-G (IgG), fibrinogen and albumin extravasation (Fig. 1e, g; Supplementary Fig. 1d) evident in each sample when compared to non-diseased tissues. Western blot analysis of hippocampal protein samples from the hippocampus of non-diseased and TLE patients revealed levels of claudin-5 protein were significantly decreased in resected tissues from the TLE patients (Supplementary Fig. 1e).

**Focal induction of seizures induces tight junction disruption and BBB leakage.** To explore whether seizure activity directly drives changes to claudin-5, we evoked seizures in male C57BL/6J mice with the excitotoxin kainic acid. Intra-hippocampal injection of 0.3 µg kainic acid for 3 h (acute) and 4 weeks (chronic) resulted in distinct behavioural abnormalities in mice that we have previously reported in claudin-5-deficient animals with increased anxiety and hyperlocomotion as assessed by distance travelled and the number of entries to the centre zone in the open field test (Fig. 2a and Supplementary Fig. 2a). Hallmark features of the model were present including acute induction of cFos, chronic upregulation of the astrocyte protein GFAP, selective neurodegeneration in the CA1 and CA3 and granule cell dispersion in the dentate gyrus (Supplementary Fig. 2b, c). We next assessed vascular patterning in the brain following local injection of kainic acid via collagen IV immunostaining in the hippocampus and surrounding regions. There were reductions in vessel coverage, number of junctions and endpoints in the CA3 and dentate gyrus of kainic acid injected mice at the chronic timepoint, with these mice also showing reduced average vessel length in the dentate gyrus. In contrast, there was increased vessel coverage, junctions and endpoints in the somatosensory cortex in kainic acid injected mice at the chronic timepoint. At the acute timepoint, kainic acid injection resulted in increased number of endpoints in the CA1, CA3 and somatosensory cortex (Supplementary Fig. 3). Similar to resected human tissue, an aberrant pattern of claudin-5 expression was observed in C57BL/6J mice receiving intrahippocampal injection of kainic acid (Fig. 2b) with the percentage area of claudin-5 also decreased along the microvessel length compared to CD31 (Fig. 2c). Transcript levels of *Cldn5* (Fig. 2d) along with other tight

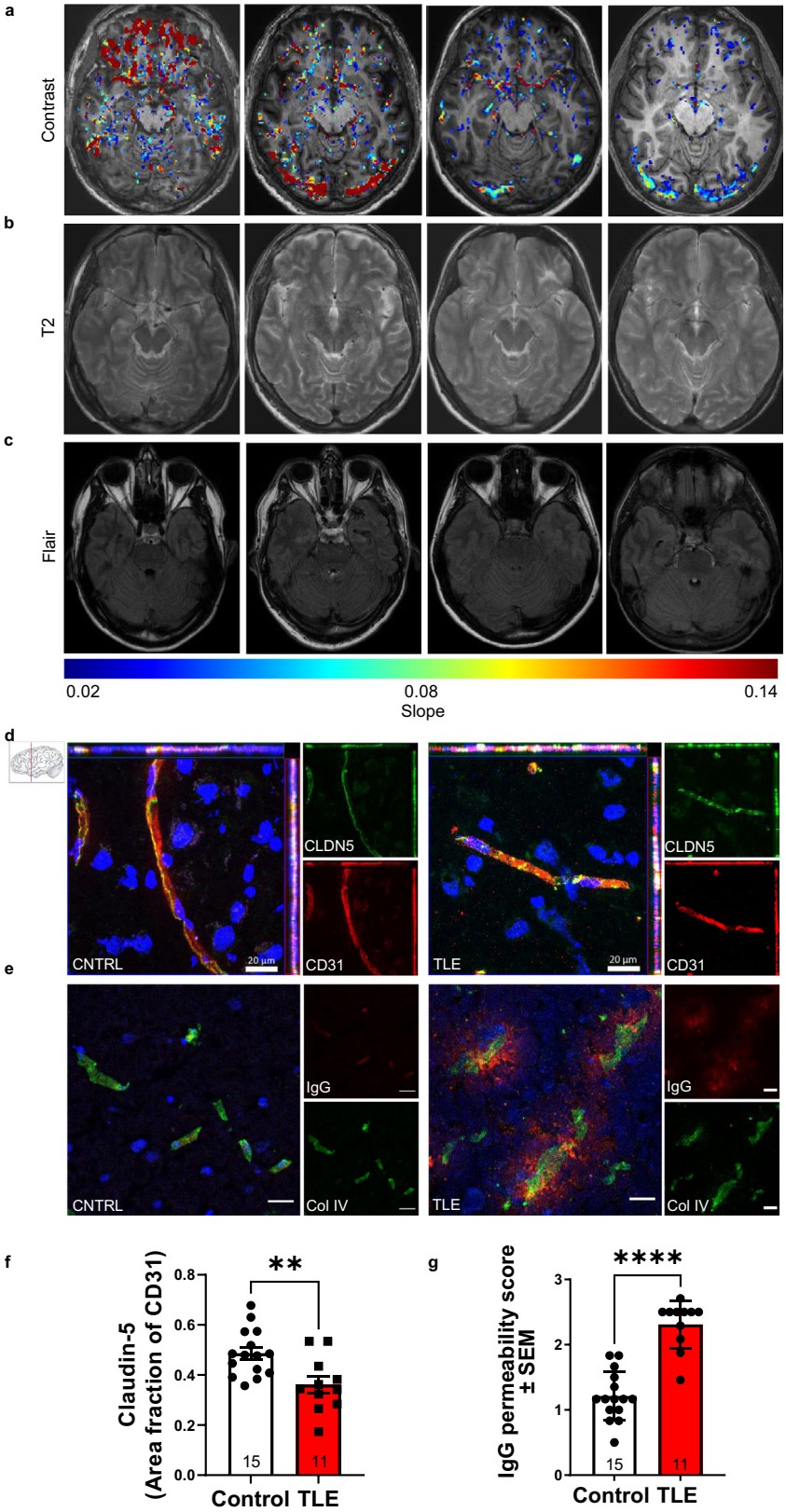

junction associated components *Ocln* (Fig. 2e) and *Tjp1* (Fig. 2f) were also significantly decreased in the hippocampus in the acute phase post kainic acid injection. We next investigated brain-blood vascular leakage by assessment of the levels of the astrocytic protein S100β which is a marker of BBB disruption and has previously been reported to be elevated in epilepsy[23]. Serum analysis of S100β showed elevated levels within 3 h of kainic acid injection and again 5 weeks later when spontaneous seizures were recurrent (Fig. 2g). Additionally, extravascular biotin was observed in the acute and chronic stages when claudin-5 levels were discontinuous along the endothelial cell margins (Fig. 2h–i).

**Fig. 1 Drug resistant temporal lobe epilepsy induces blood-brain barrier leakage and *Cldn5* downregulation in the hippocampus. a** Representative dynamic contrast-enhanced MRI (DCE-MRI) images from 4 patients with temporal lobe epilepsy (TLE, top panel). Colour bar represents slope of contrast agent. **b** T2 weighted images (middle panel) and (**c**) fluid attenuated inversion recovery (FLAIR) images (bottom panel). **d** Claudin-5 (CLDN5; green) and CD31 (red) staining in non-diseased autopsy control (CNTRL) and TLE cases. Scale bars, 20 µm. **e** Collagen-IV (ColIV; green) and IgG (red) staining in CNTRL and TLE cases. Scale bars, 50 µm. **f** Levels of CLDN5 were significantly decreased (**$p = 0.0042$) and **g** IgG was significantly increased (****$p < 0.0001$) in the brains of TLE cases vs CNTRL. Data represent means ± s.e.m.; each datapoint represents one patient. Number of subjects (*n*) is indicated on graphs. Two-sided Mann–Whitney test for immunohistochemical and western blot analysis. **$p < 0.01$; ****$p < 0.0001$. Source data are provided as a Source Data file.

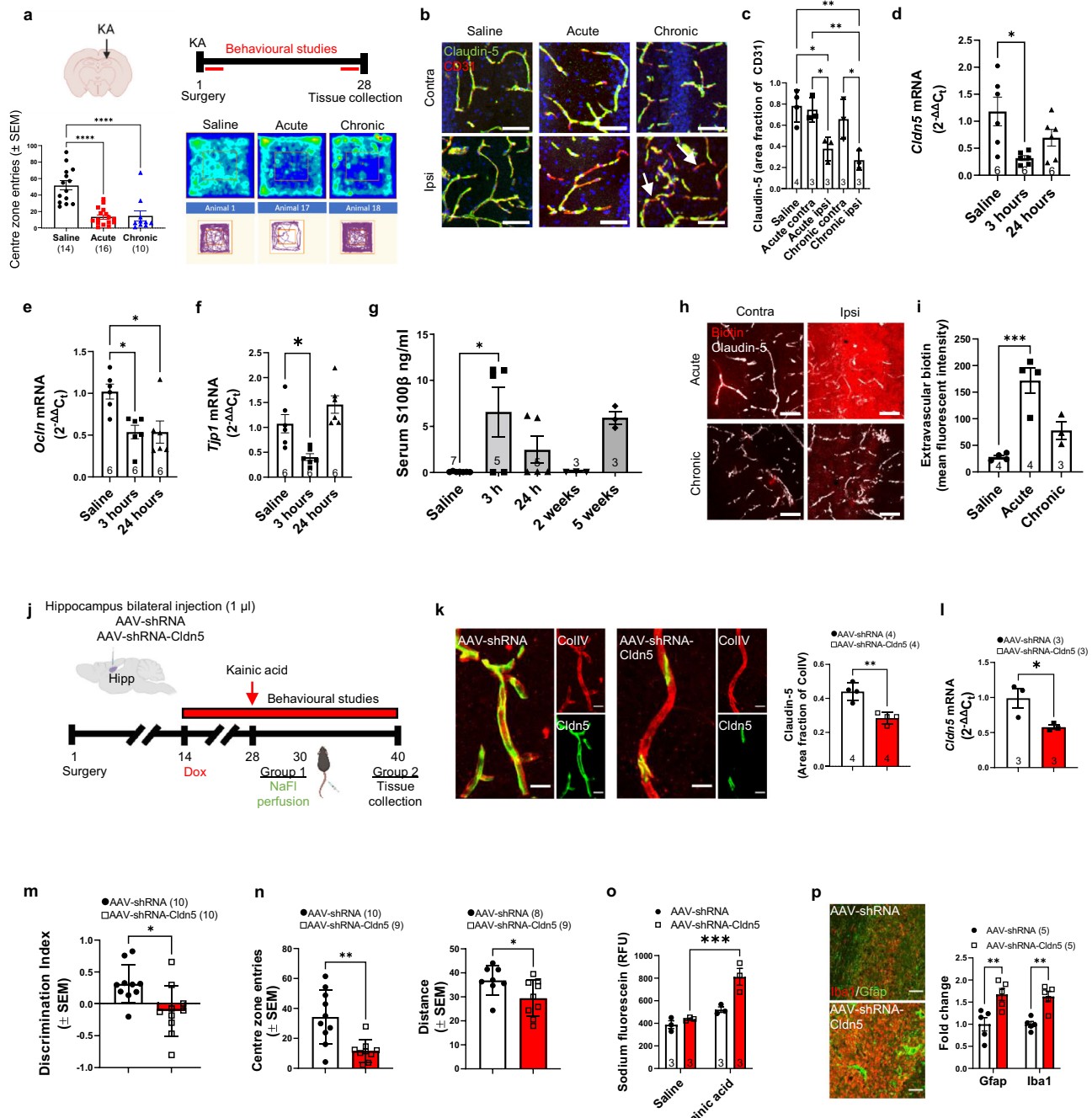

**Targeted downregulation of claudin-5 in the hippocampus exacerbates kainic acid induced seizures.** While kainic acid injection could induce a decrease in claudin-5 levels at the BBB, in an effort to specifically target claudin-5, male C57BL/6J mice were stereotaxically injected into the dorsal hippocampus with a doxycycline inducible adeno-associated virus (AAV2/9) vector expressing claudin-5 shRNA (AAV-shRNA-Cldn5) or a non-targeting control AAV2/9 (AAV-shRNA) (Fig. 2j). Mice were supplemented with doxycycline in their drinking water for three weeks to downregulate claudin-5 protein and mRNA specifically

**Fig. 2 Conditional knockdown of *Cldn5* expression in the hippocampus exacerbates kainic acid-induced epilepsy. a** Kainic acid (KA) injection site and timeline of experiment (top). Figure Created with BioRender. Open field testing of mice post KA injection with significant decreases in the number of entries in the centre zone (bottom) at both timepoints compared to saline (****$p < 0.0001$ for acute and chronic). **b** Claudin-5 (green) and CD31 (red) staining in the hippocampus in saline injected animals and the acute and chronic stages post KA injection. Scale bars, 50 μm. **c** Levels of claudin-5 expression were significantly decreased following KA injection at the acute (*$p = 0.0187$) and chronic stage (**$p = 0.0036$). **d** *Cldn5* (*$p = 0.0102$), **e** *Ocln* (*$p = 0.0124$) and **f** *Tjp1* (*$p = 0.0129$) mRNA levels were decreased at 3 h and 24 h post KA injection. **g** S100B levels were significantly increased in serum of mice 3 h post KA injection (*$p = 0.011$). **h** Immunohistochemistry for Sulfo-NHS-Biotin and claudin-5 following intrahippocampal injection of KA. Scale bars, 50 μm. **i** Biotin extravasated into the brain at the acute stage post injection of KA (***$p = 0.0007$). Figure Created with BioRender. **j** AAV expressing claudin-5 shRNA injected bilaterally into the hippocampus and timeline of experiment. **k** Claudin-5 protein (**$p = 0.0024$) (green) and **l** mRNA (*$p = 0.043$) was significantly decreased in brain endothelial cells (red) following AAV-shRNA-Cldn5 injection in the hippocampus of mice compared to that in AAV-shRNA-injected mice, following doxycycline (Dox) treatment. Scale bar, 20 μm. **m** Reduced performance in the novel object recognition (*$p = 0.0146$) and **n** open field test (**$p = 0.0023$) post claudin-5 suppression in KA injected mice. **o** Increased sodium fluorescein (NaFl) extravasation post claudin-5 suppression in KA injected mice (***$p = 0.0006$). **p** Increased IBA1 positive microglia (**$p = 0.0044$) and GFAP positive astrocytes (**$p = 0.0024$) post claudin-5 suppression in kainic acid injected mice. Scale bar, 20 μm. Data represent means ± s.e.m.; each datapoint represents one animal. Number of animals (*n*) is indicated on graphs. Two-way ANOVA followed by Bonferroni's multiple comparison test for sodium fluorescein permeability; two-sided unpaired *t*-test for AAV experiments; one-way ANOVA followed by Tukey's multiple comparison test for all other graphs. *$p < 0.05$; **$p < 0.01$; ***$p < 0.001$; ****$p < 0.0001$. Source data are provided as a Source Data file.

in the dorsal hippocampus of AAV-shRNA-Cldn5 injected mice (Fig. 2k, l). Additionally, AAV-shRNA-Cldn5 injected mice had biotin extravasation in the hippocampus (Supplementary Fig. 4). AAV-shRNA-Cldn5 injected mice with focal downregulation of claudin-5 in the dorsal hippocampus, subjected to seizures by intraperitoneal (i.p.) injection of a mildly convulsant dose of kainic acid, performed worse in the novel object recognition task and had reduced distance travelled and centre zone entries in the open field test, readouts of cognitive function and anxiety respectively (Fig. 2m, n). We next investigated the extent of BBB permeability in this model. 48 h following kainic acid injection, mice were intravenously injected with 200 mg/kg sodium fluorescein and 10 min later were perfused with PBS, hippocampus were dissected and homogenised in 1 % triton-x100 and read on a fluorescent plate reader. AAV-shRNA-Cldn5 mice had greater extravasation of sodium fluorescein in the hippocampus compared to AAV-shRNA injected mice (Fig. 2o). AAV-shRNA-Cldn5 mice also showed increased neuroinflammation as assessed by Iba1 (microglial marker) and GFAP (astrocyte marker) immunoreactivity in the hippocampus CA1 (Fig. 2p).

**Claudin-5 heterozygosity makes the BBB susceptible to mildly convulsive kainic acid injection**. We next assessed seizure susceptibility in male wild-type (WT) or endothelial specific *Cldn5+/−* mice (Fig. 3a). These mice had 50 % less *Cldn5* mRNA (Fig. 3b, c) and protein (Fig. 3d) with no change in levels of *Ocln*, *Tjp1*, *Tjp2*, *Cdh5* or *Jam2* (Supplementary Fig. 5a). These mice behaved normally except for producing more errors in the y-maze (Supplementary Fig. 5b). Next, we implanted WT and claudin-5 heterozygous mice with cortical electrodes and recorded baseline EEG activity for 1 h. To test if loss of one copy of *Cldn5* increased the vulnerability to seizures, mice were then injected with a mildly convulsant dose of kainic acid and EEG activity was recorded for a further 3 h (Fig. 3e). Mildly convulsant kainic acid increased *Fos* expression 3 h post-injection without altering the expression of inflammatory markers or glutamate transporters (Supplementary Fig. 5c). EEG analysis revealed that mice heterozygous for *Cldn5* had increased EEG total power prior to kainic acid injection (Fig. 3f). Following kainic acid injection, *Cldn5* heterozygous mice had shorter latency to seizure onset (Fig. 3g), while 100% (5/5) of heterozygous mice entered status epilepticus, defined as more than 5 min of continuous seizures, compared to 16% (1/6) of WT mice (Fig. 3h). *Cldn5* heterozygous mice had increased total power and spent more time in ictal activity following kainic acid injection (Fig. 3i, j). Representative EEG tracings show distinct seizure activity in *Cldn5* heterozygous mice following

injection of mildly convulsant kainic acid (Fig. 3k). In a separate cohort of WT or *Cldn5* heterozygous mice, we scored mice according to the Racine scale following injection of a mildly convulsant dose of kainic acid. Mice heterozygous for *Cldn5* had greater cumulative seizure score (Fig. 3l), shorter latency to seizure onset (Fig. 3m) and greater seizure severity (Fig. 3n). Furthermore, these mice had more CNS inflammation as assessed by immunoreactivity for GFAP and IBA1 (Fig. 3o). Together, these findings indicate that lowering claudin-5 levels increases vulnerability to evoked seizures, disrupts BBB integrity and promotes neuroinflammation.

**Spontaneous recurrent seizures in claudin-5 knockdown mice**. Previously, we reported on the development of a doxycycline-inducible claudin-5 knockdown mouse model which allows for constitutive knockdown of *Cldn5* mRNA and protein into adulthood (Fig. 4a, b). We showed that these mice die after several weeks of doxycycline treatment and display several learning and memory and anxiety/depression deficits[22]. During the period of continuous claudin-5 depletion, mice also display evidence of spontaneous recurrent seizures (SRS), (Fig. 4c). Continuous EEG recording with simultaneous video recording in male mice revealed ictal EEG activity within 5 weeks of doxycycline administration with Racine stage 5 seizures (Fig. 4d). These data suggest lowering claudin-5 levels is sufficient to cause epilepsy in mice. As prolonged claudin-5 suppression is lethal in mice, we designed an experiment that would allow for recovery of claudin-5 levels (Fig. 4f) in an effort to explore if neural damage could be reversed. Indeed, prolonged claudin-5 suppression caused an increase in hyper-proliferation of Iba1-positive microglia (Fig. 4e). In the rescue experiment where mice were supplemented with doxycycline for 4 weeks and then returned to normal drinking water for a further 2 weeks, seizure behaviour and astrogliosis as evidenced by GFAP positivity were significantly reversed (Fig. 4f, g).

**Claudin-5 knockdown activates endothelial cells**. In an effort to understand the role of claudin-5 levels in the maintenance of endothelial cell integrity, siRNA targeting *Cldn5* mRNA was transfected into mouse brain endothelial cells and cell proliferation, motility and inflammation was assessed. Claudin-5 siRNA transfected cells had significantly less claudin-5 protein (Supplementary Fig. 6a) and displayed reduced rates of proliferation at 24 h and 5 days post suppression (Supplementary Fig. 6b). Claudin-5 knockdown enhanced endothelial motility in the

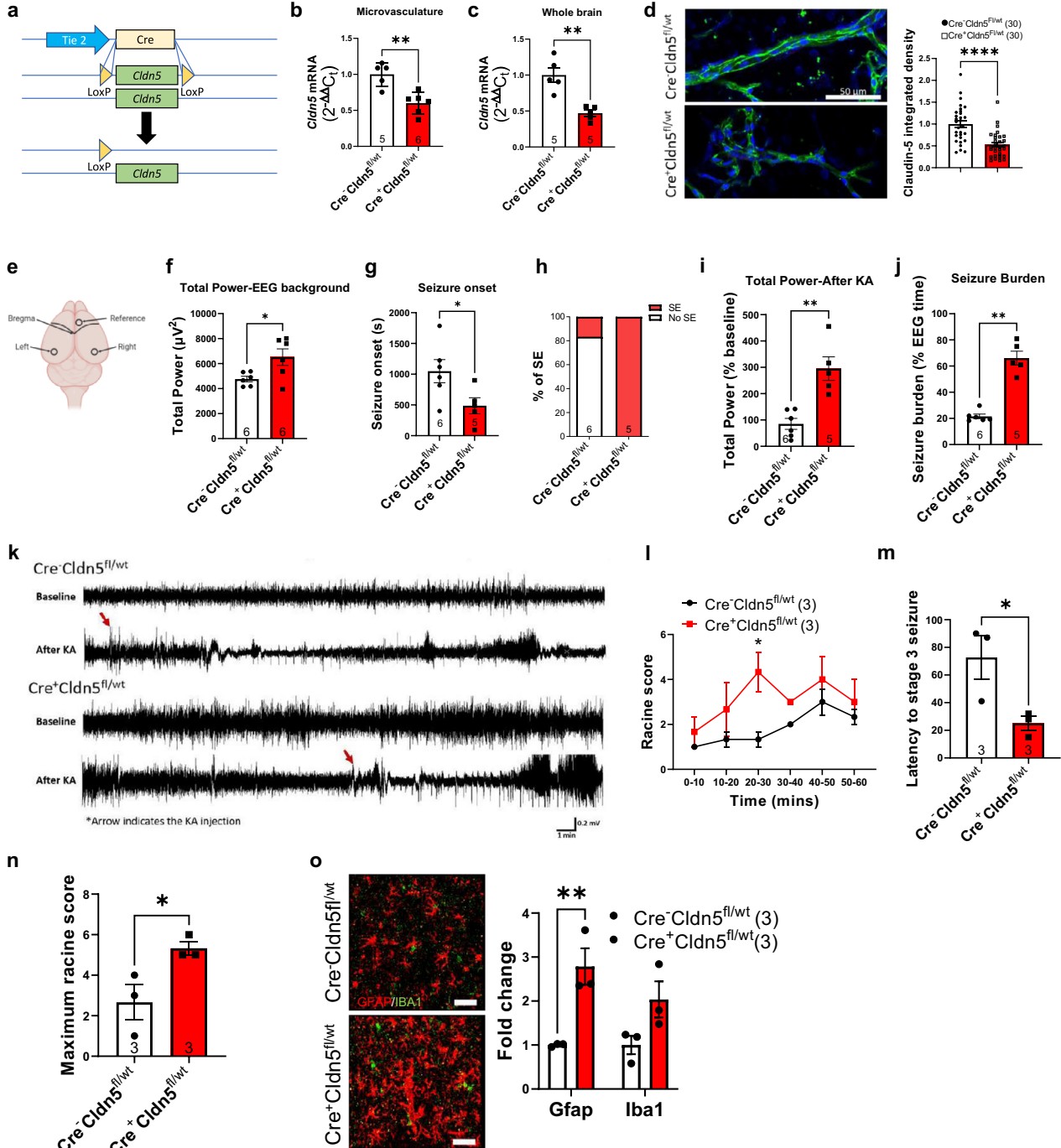

**Fig. 3 Cldn5 heterozygous mice have reduced threshold for kainic acid-evoked seizures. a** Schematic overview of *Cldn5fl/wt* mouse. **b** *Cldn5* transcript levels were significantly decreased *Cre+Cldn5fl/wt* mice in isolated microvasculature (**$p = 0.0023$) and **c** whole brain (**$p = 0.0014$). **d** Claudin-5 protein expression significantly decreased in isolated microvessels of *Cre+Cldn5fl/wt* mice (****$p < 0.0001$). Scale bar, 50 µm. **e** Schematic representation of electrode placement for electroencephalography (EEG) studies (top). **f** Increased total power in *Cre+Cldn5fl/wt* mice prior to kainic acid (KA) injection (*$p = 0.0311$). **g** Reduced latency to the first electrographic seizure (*$p = 0.0432$) in *Cre+Cldn5fl/wt* mice. **h** Percentage of mice entering status epilepticus (SE) following kainic acid injection. **i** Increased total power in *Cre+Cldn5fl/wt* mice following mildly convulsant KA injection (**$p = 0.0015$). **j** Increased seizure burden in *Cre+Cldn5fl/wt* mice following KA injection (**$p = 0.0043$). **k** Representative EEG recording post mildly convulsive KA injection. **l** Increased Racine score over time in *Cre+Cldn5fl/wt* mice (*$p = 0.0148$ at 20–30 min post KA injection). **m** Decreased latency to stage 3 seizures in *Cre+Cldn5fl/wt* mice (*$= 0.0459$). **n** Increased racine score in *Cre+Cldn5fl/wt* mice post injection of KA (*$p = 0.0474$). **o** Increased GFAP (red) expression in *Cre+Cldn5fl/wt* mice (**$p = 0.0072$). IBA1 (green) expression did not change ($p = 0.0894$). Scale bars, 50 µm. Data represent means ± s.e.m.; each data point represents one animal or vessel. Number of animals (*n*) is indicated on graphs. Two-way ANOVA followed by Bonferroni's multiple comparison test for Racine score over time; two-sided Mann–Whitney test for seizure burden; two-sided unpaired *t*-test for all other graphs. *$p < 0.05$; **$p < 0.01$; ****$p < 0.0001$. Source data are provided as a Source Data file.

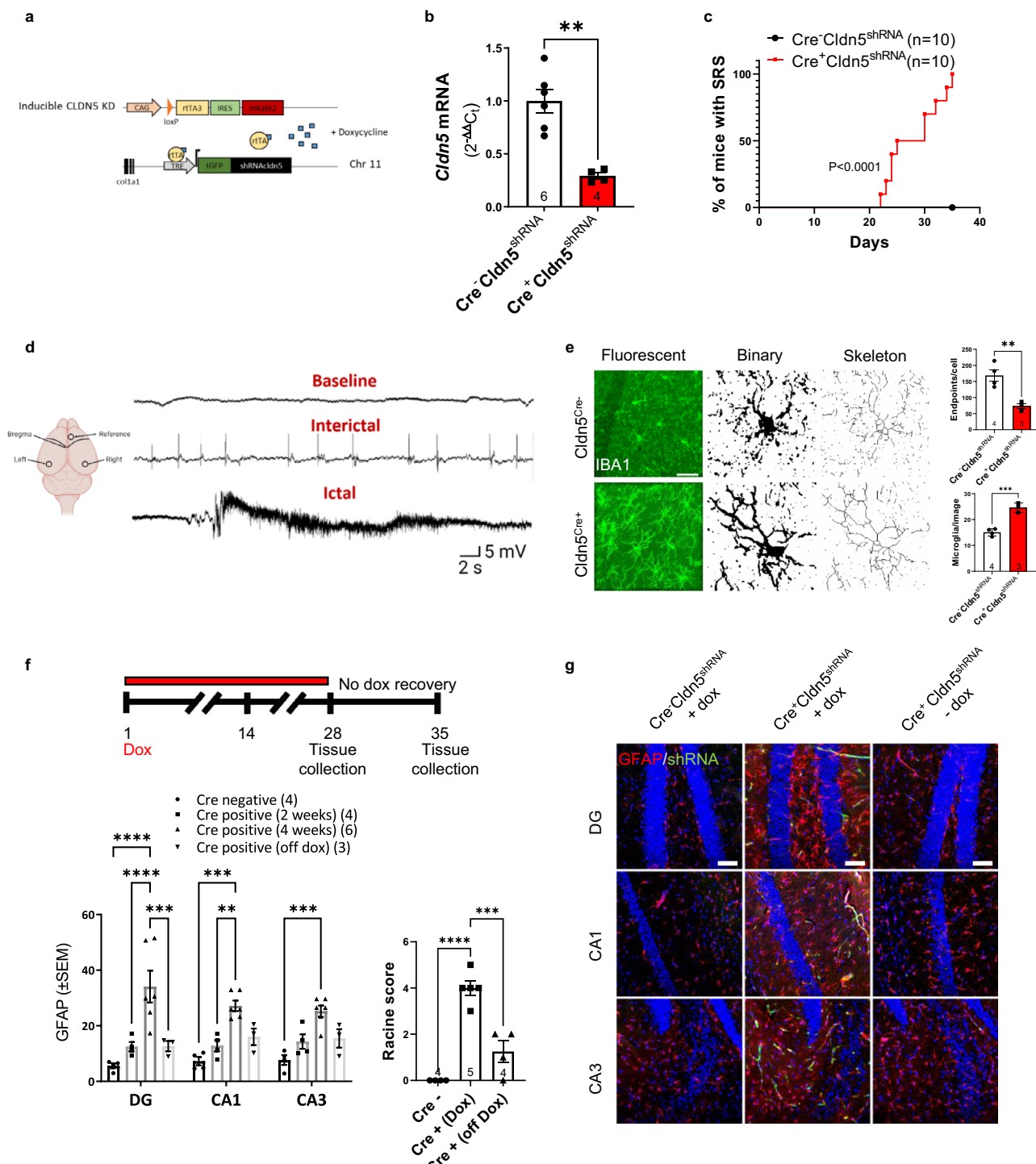

**Fig. 4 Inducible *Cldn5* knockdown mice develop spontaneous seizures and neuroinflammation. a** Schematic outline of inducible knockdown mouse model. **b** *Cldn5* significantly suppressed following 4 weeks of doxycycline (Dox) treatment (**$p = 0.001$). **c** Percentage of mice with spontaneous recurrent seizures (SRS) following 4 weeks of Dox treatment (****$p < 0.0001$). **d** Representative EEG traces following 4 weeks of Dox treatment. **e** IBA1 (green) quantification post claudin-5 suppression with decreased cellular endpoints (**$p = 0.0088$), and increased IBA1 positive cells (***$p = 0.0005$). Scale bar, 50 μm. **f, g** Experimental outline for recovery experiment. Increased GFAP levels in the dentate gyrus (DG ****$p < 0.0001$), CA1 (***$p = 0.0001$) and CA3 (***$p = 0.0007$) of mice following 4 weeks of doxycycline treatment are reversed following doxycycline withdrawal in the DG (***$p = 0.0002$ vs Cre positive 4 weeks). Scale bar, 100 μm. Increased seizure behaviour as scored according to Racine scale (****$p < 0.0001$ for Cre− vs Cre+ (Dox)) decreased following Dox removal (***$p = 0.0004$ for Cre+ (Dox) vs Cre+ (off Dox). Data represent means ± s.e.m.; each datapoint represents one animal. Number of animals (*n*) is indicated on graphs. Mantel–Cox test for SRS analysis; two-sided unpaired *t*-test for qPCR and microglia morphology; two-way ANOVA followed by Bonferroni's multiple comparison test for reversibility study. **$p < 0.01$; ***$p < 0.001$; ****$p < 0.0001$. Source data are provided as a Source Data file.

scratch assay (Supplementary Fig. 6c, d). Previous studies have identified ICAM-1 as a positive regulator of cellular motility[24] and qPCR analysis revealed that claudin-5 depletion induced expression of *Icam1* in a dose-dependent manner and this was further enhanced by pre-treatment with pro-inflammatory mediator IL-1β. Signalling pathways related to transmigration were also activated following knockdown of claudin-5, including, *Vcam1* and *Sele* (Supplementary Fig. 6e–i). Finally, Western blot, qPCR and immunohistochemical analysis of hippocampal samples from male WT or *Cldn5* heterozygous mice revealed a significant increase in ICAM-1 expression as well as the appearance of ICAM-1 positive infiltrating cells in claudin-5 deficient mice injected with a mildly convulsant dose of kainic acid (Supplementary Fig. 6j–l). Together, this data suggests that downregulation of claudin-5 results in the upregulation of cell adhesion molecules and increased immune cell infiltration following a subconvulsant injection of kainic acid.

**Stabilisation of BBB integrity attenuates seizures and decreases neural damage in kainic acid-induced epilepsy.** Previous studies have highlighted the role of transforming growth factor beta (TGFβ) in the development of epilepsy[14]. We performed qPCR for markers of the TGFβ signalling pathway in human TLE resected samples. This revealed upregulation of TGFβ1, SMAD2/3 and SNAI1 mRNAs (Supplementary Fig. 7a). Additionally, it has been shown that targeting the ALK5 signalling component of the TGFβ pathway can regulate levels of claudin-5 while a recent study identified RepSox, a potent ALK5 inhibitor, as a potent regulator of claudin-5 expression[25]. We confirmed that 24-h treatment with RepSox could induce *Cldn5* mRNA expression in hCMEC/d3 cells (Fig. 5a), without affecting other junctional components including occludin, tricellulin, ZO-1 and ZO-2 (Supplementary Fig. 7b). This increase was associated with an increase in barrier properties as revealed by increased transendothelial electrical resistance (TEER) (Fig. 5b) and reduced permeability to 40 kDa FITC-dextran (FD40) 24 h post treatment with RepSox (Fig. 5c and Supplementary Fig. 7c). RepSox treatment also attenuated TGFβ and VEGF-induced flux of FD40 in hCMEC/d3 cells (Fig. 5d) and rescued TGFβ-induced loss of CLDN5, OCLN and TJP2 mRNA (Fig. 5e–g). Next, we investigated the therapeutic effect of RepSox on the acute effects of intrahippocampal kainic-acid induced BBB disruption in male C57BL/6J mice (Fig. 5h). Two 10 mg/kg i.p. injections of RepSox was sufficient to reduce evoked seizures when mice were scored according to the Racine scale (Fig. 5i). In agreement with this, RepSox reduced neural activity when assessed by cFos immunoreactivity with reductions in the cortex and CA3 region of the hippocampus in mice receiving a unilateral kainic acid injection in the dorsal hippocampus (Fig. 5j). At the acute stage, RepSox significantly attenuated BBB disruption as assessed by leakage of sodium fluorescein and IgG (Fig. 5k). We also assessed the effect of RepSox at 7 days post kainic acid injection. RepSox significantly reduced astrogliosis and attenuated the extravasation of IgG (Supplementary Fig. 7d, e). Finally, RepSox significantly rescued kainic acid-induced deficits in claudin-5 expression and localisation (Supplementary Fig. 7f). Together, we show that RepSox can stabilise BBB integrity and rescue neural damage associated with kainic acid in part due to restoration of claudin-5 along the cerebral vasculature.

## Discussion
Overall, our findings suggest that epilepsy alters BBB integrity through modulation of the tight junction protein claudin-5 which facilitates a local inflammatory response and the passage of blood-derived proteins and immune cell infiltration into the brain parenchyma. While loss of claudin-5 is lethal with severe neuroinflammation and recurrent seizures occurring, haploinsufficiency is sufficient to reduce the threshold for induction of convulsive seizures with more severe seizures and reduced latency to seizures evident. This complements decades of research showing that BBB breakdown is sufficient to induce epileptiform activity[6–8,14,26,27]. The diffuse and discontinuous claudin-5 patterns of immunoreactivity we observed in resected hippocampus and cortex in the temporal lobe of patients with epilepsy suggests BBB dysfunction and claudin-5 dysregulation are key drivers of pathology. Added to this, the extensive gadolinium extravasation assessed by DCE-MRI suggests that BBB disruption propagates seizure activity and could contribute to pharmaco-resistance in patients. Indeed, the pattern of claudin-5 expression in resected tissues suggest that the BBB is hyper-permeable in these regions and blood-borne material can easily diffuse into the delicate neural tissues. In this regard, it has been suggested that localised BBB permeability may contribute to epilepsy which is refractory to medication due to reduced bioavailability of anti-seizure drugs[28]. In the mouse model of TLE, we observed acute downregulation of claudin-5 protein and mRNA with levels increasing during the chronic epileptic phase. This may suggest that dysregulation of tight junction structure and function occurs early during the processes of epileptogenesis. It is also plausible that the re-constituted tight junction is then unable to reach a critical threshold to maintain homoeostasis and becomes susceptible to further breakdown with much lower stimuli. These findings are corroborated by our transgenic studies. Localised AAV-mediated knockdown of claudin-5 in the dorsal hippocampus exacerbated cognitive decline observed in the kainic acid model of TLE while it also increased BBB permeability and neuroinflammation, suggesting that claudin-5 may help to maintain normal neural function by regulating the cerebral vasculature and the local inflammatory response. In this regard, a recent study has shown that the methylation pattern of the claudin-5 locus is associated with cognitive decline even in individuals with little or no sign of amyloid or neurofibrillary tangle pathology, highlighting the importance of normal BBB function in aging and that claudin-5 may be vital to these processes[29]. Chronic suppression of claudin-5 in the central nervous system (CNS) induced spontaneous seizures in mice with severe neuroinflammation. Added to this, claudin-5 haploinsufficiency was sufficient to exacerbate the acute effects of mildly convulsive kainic acid on seizure severity, cognitive function, BBB integrity and neuroinflammation. Mechanistically, we have shown that the loss of claudin-5 can facilitate an activated endothelial cell fate with upregulation of cell adhesion molecules. This is important, as there are a plethora of studies showing increased local inflammatory status of resected tissues from patients with epilepsy with evidence for infiltrating T cells in intractable paediatric epilepsy[30]. In experimental models, status epilepticus was associated with increased infiltration of CCR2 expressing monocytes and blockade of these blood-derived monocytes was protective[31]. We propose that this local inflammation is both a cause and result of BBB disruption. Loss of claudin-5 and disrupted BBB integrity likely results in a release of chemoattractants into the circulation resulting in the recruitment of immune cells. However, our in vitro studies reveal that loss of claudin-5 alone can facilitate immune-vascular interactions by inducing the expression of cell adhesion molecules such as ICAM-1 and VCAM-1 which have previously been shown to be proepileptogenic[32].

Intriguingly, we show that kainic acid induces downregulation of claudin-5 and subsequent BBB breakdown but also that claudin-5 knockdown/knockout triggers seizure behaviour by itself. This suggests the possibility of a positive feedback loop where a predisposed vulnerable BBB leads to the development of

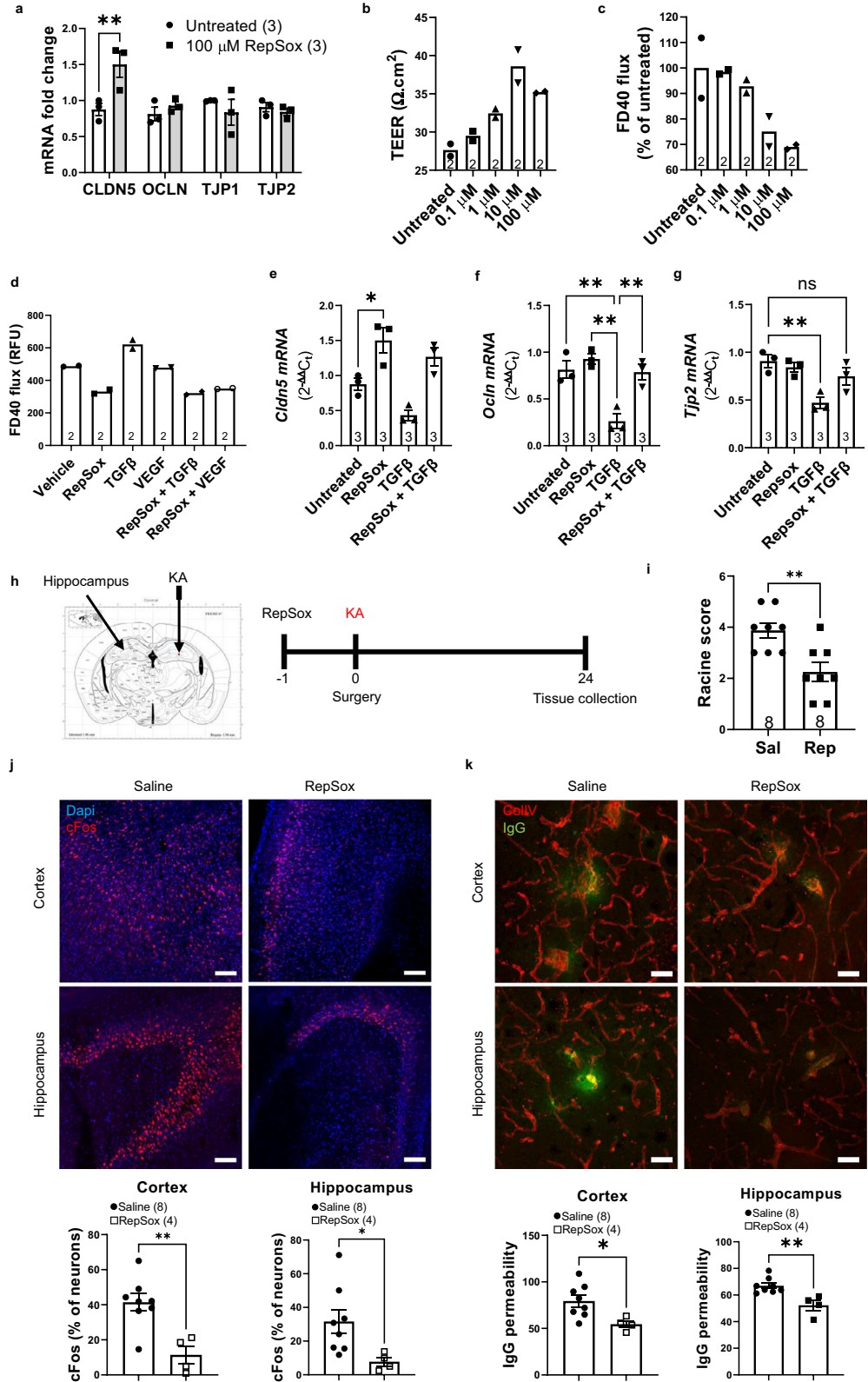

seizures which subsequently leads to sustained BBB permeability. It has been shown by others that BBB permeability can lead to seizures and the degree of BBB breakdown positively correlates with the severity of seizures. This is also evident in other models such as the chronic social defeat stress model of depression which is associated with a decrease in claudin-5 expression in the nucleus accumbens while viral-mediated downregulation of

claudin-5 is sufficient to induce depressive-like behaviours[33]. While many studies have shown that epilepsy is associated with a breakdown of the BBB, it is also important to recognise that enhancement of BBB properties may also be involved in the development of pharmacoresistance in epilepsy. Drugs designed to act on the brain have been made lipophilic to readily diffuse across the plasma membrane of endothelial cells in the brain,

**Fig. 5 ALK5 inhibition induces *Cldn5* expression and attenuates kainic acid-induced seizures and BBB leakiness. a** Increased *Cldn5* expression 24 h post treatment with RepSox (100 µM) (**$p = 0.0029$). **b** Increased trans-endothelial electrical resistance (TEER) with increasing doses of RepSox. **c** Decreased flux of FITC-Dextran-40 (FD40) with increasing doses of RepSox. **d** Decreased flux of FD40 following exposure of endothelial cells to transforming growth factor-β1 (TGFβ) or vascular endothelial growth factor (VEGF) in combination with 100 µM RepSox. **e** RepSox significantly increased *Cldn5* mRNA (*$p = 0.0182$) without changing (**f**) *Ocln* and (**g**) *Tjp2* expression. **h** Experimental outline of RepSox treatment in the intrahippocampal kainic acid (KA) induced seizure model. **i** Reduced Racine score in RepSox treated mice (**$p = 0.0039$) compared to saline (Sal). **j** Decreased cFos expression in the cortex (**$p = 0.0036$) and hippocampus (*$p = 0.0461$) post treatment with 10 mg/kg RepSox. Scale bar, 100 µm. **k** Decreased IgG (green) extravasation in the cortex (*$p = 0.0243$) and hippocampus (**$p = 0.0038$) post treatment with 10 mg/kg RepSox. Scale bar, 100 µm. Data represent means ± s.e.m.; number of independent experiments or animals (*n*) is indicated on graphs. Two-way ANOVA followed by Bonferroni's multiple comparison test for tight junction qPCRs; one-way ANOVA followed by Tukey's multiple comparison test for cell culture experiments; two-sided unpaired *t*-test for in vivo studies. *$p < 0.05$; **$p < 0.01$; ****$p < 0.0001$. Source data are provided as a Source Data file.

however this makes them potential substrates for efflux carriers at the BBB, especially P-glycoprotein (Pgp). Many studies have indicated that in conditions with a disrupted BBB, including epilepsy, defense mechanisms in brain capillary endothelial cells and perivascular glia may be upregulated, including an increased expression and functionality of Pgp and other efflux carriers and drug-metabolising enzymes[34]. In resected brain tissue from pharmacoresistant epilepsy patients, an elevated level of drug efflux transporters including Pgp and BCRP was observed. Additionally, elevated levels and activity of cytochrome P450 enzymes was found in epileptic brain tissue[11]. Animal studies have revealed that seizures induce expression of Pgp while rats with spontaneous seizures not responding to anti-seizure drugs (ASD) show higher expression of Pgp at the BBB compared to rats responsive to ASDs. Furthermore, animals treated with the Pgp inhibitor tariquidar can be converted to ASD responders[35–38]. This hypothesis of BBB-associated drug resistance in epilepsy has also been investigated clinically using positron emission tomography (PET) with the PET ligand and Pgp substrate verapamil with and without the Pgp inhibitor tariquidar. In 14 pharmacoresistant TLE patients, 8 seizure free patients and 13 healthy controls, higher Pgp functionality was found in the pharmacoresistant patients compared to seizure-free patients in several brain regions. Higher Pgp activity was associated with higher seizure frequency[39]. Together, this has been suggested to contribute to ASD resistance in epilepsy which affects about 30% of all patients.

It is becoming clear that claudin-5 participates in other endothelial cell functions and is not solely involved in barrier formation. Indeed, previous studies have shown that loss of claudin-5 can influence endothelial cell migration and proliferation as well as being of paramount importance for maintaining the restrictive paracellular permeability properties of the BBB[21,40–42]. We complement these studies by showing that loss of claudin-5 enhances endothelial cell migration while inhibiting proliferation. Furthermore, loss of claudin-5 influences endothelial cell activation by upregulating vascular adhesion molecules following IL-1β treatment. Interestingly, the role of claudin-5 in immune cell trafficking at vascular barriers has been understudied. Recently, however, it was shown that thymocyte trafficking preferentially occurs at claudin-5 negative blood vessels in the thymus[43]. In another study, the appearance of claudin-5 positive leucocytes around inflamed vessels in experimental autoimmune encephalomyelitis suggests that claudin-5 transfer from the endothelium to circulating leucocytes may facilitate immune cell transmigration[44]. Collectively, our findings suggest that loss of BBB integrity can lead to the development of epilepsy. Therapies that regulate BBB integrity may represent a robust way of treating patients who become pharmaco-resistant to the current standard of care. BBB regulating drugs such as RepSox and others, could also potentially be used in combination with current anti-seizure medications. Additionally, targeted molecular or gene

therapy-based approaches directly targeting claudin-5 could pave the way for a new generation of drugs to treat epilepsy and other neurological conditions where BBB breakdown is a hallmark pathology. How claudin-5 is modulated in the epileptic brain is currently unknown and requires further studies. However, recent work has shown that neural activity can modulate key properties of the BBB including the expression of tight junction components[45]. Claudin-5 has been found to be dysfunctional in many neurological disorders[40], but the effects of claudin-5 upregulation have not been studied. Based on our findings, the next logical step will be to assess the therapeutic efficacy of modulating claudin-5 in epilepsy models via small molecule regulation or gene therapy delivery of claudin-5 cDNA. If this proves beneficial in rodent models of epilepsy, then the therapy could be trialled in non-human primates prior to clinical deployment.

## Methods

**Study participants**. One healthy and 4 temporal lobe epilepsy (TLE) volunteers were recruited from Dublin, Ireland, and informed consent was obtained from all participants. Resected brain tissue was obtained from 16 TLE patients and 11 autopsy control samples from the Stanley Medical Research Institute. The St James' Hospital ethics committee approved the human studies and informed consent was obtained from all participants. Research was performed according to the principles of the Declaration of Helsinki.

**Mice and genotyping**. All animal experiments were performed in accordance with the European Communities Council Directive (2010/63/EU), the NIH Guide for the Care and Use of Laboratory Animals and followed ARRIVE guidelines. Protocols were reviewed and approved by the Research Ethics Committees of Trinity College Dublin and/or the Royal College of Surgeons in Ireland under licenses from the Department of Health (HPRA: AE19136/PO80 and/or AE19127/P057), Dublin, Ireland. Mice were housed under a 12-h light/dark cycle with temperature and humidity maintained at 18–23 °C and 40–50% respectively. Mice were provided ad libitum access to standard food and water. Animals were acclimated to their environment for 1 week prior to experimental procedures. Several lines of mice were used for experiments. Cldn5-GFP (GFP expressed under the control of claudin-5 promoter) were generated in the Department of Immunology, Genetics, and Pathology, Rudbeck Laboratory, Uppsala University, Sweden[46]. Tie2-cre (CRE expressed under Tie2 promoter) were crossed to Cldn5fl/fl mice for endothelial specific deletion of claudin-5 (Vazquez et al., manuscript in preparation). Doxycycline-inducible Cldn5shRNA mice were described previously[22]. All mice were bred on-site at TCD onto a C57BL/6J background and used at 8–14 weeks of age for all experiments.

**Seizure induction and scoring**. All mice were bred on-site in the specific pathogen-free unit at the Smurfit Institute of Genetics in TCD. Adult male (~20 g), C57BL/6J mice (8–12 weeks old) were anaesthetised with a mixture of ketamine/ medetomidine (100/0.25 mg/kg) and placed in a stereotaxic frame. A 1 cm midline incision was made in the scalp and a burr hole was made at A/P −2 mm; M/L −1.5 mm; D/V −2 mm. A Hamilton syringe was slowly lowered into the dorsal hippocampus and 0.1 µL of a 10 mM solution of kainic acid was injected at a rate of 0.1 µL per minute. Afterwards, the needle was kept in place for 2 min. Mice were sacrificed at 3 h, 48 h and following the development of spontaneous recurrent seizures (2–5 weeks). In some strains, seizures were induced by an i.p injection of 15 mg/kg kainic acid. Seizure severity was recorded according to the Racine scale: 0: no change in behaviour; 1: immobility; 2: head nodding; 3: forelimb clonus; 4: forelimb clonus with rearing and falling; 5: wild running and jumping often followed by generalised tonic-clonic activity and loss of postural tone. Mice were

videotaped for 2 h following kainic acid injection and were scored by an individual blind to genotype and/or treatment[47]. Serum samples were collected following each time point for quantification of S100β.

**Stereotaxic delivery of AAV2/9**. Male, 8-14 week old C57/BL6J mice were anaesthetised and set up in a stereotaxic frame as for seizure induction. A Hamilton syringe was loaded with an AAV2/9 expressing either a short hairpin RNA (shRNA) against claudin-5 (AAV-shRNA-Cldn5) or a non-targeting control (AAV-shRNA)[48] and the needle was slowly lowered into the dorsal hippocampus. 0.5 μL of the AAV solution was then injected at a rate of 0.2 μl/minute and once complete, the needle was left in place for 2 min before repeating the procedure in the other hemisphere. Mice were given 7 days of recovery before introducing doxycycline into their drinking water (2 mg/mL in 2% sucrose solution). 2 weeks after doxycycline treatment mice were injected i.p. with 15 mg/kg kainic acid. 48 h later mice were injected with 200 mg/kg sodium fluorescein for 10 min, anaesthetised and perfused with 10 ml PBS. The brain was removed, hippocampus and cortex dissected and homogenised in 1 ml of 1% Triton X-100 in PBS. Homogenates were spun at 18,000 g for 10 min at 4 °C and supernatants were removed for fluorescence measurements. For behavioural experiments, mice were assessed by the novel object recognition and open field test as described below.

**Open field test**. Mice were handled for approximately 5 min per day for 1 week prior to behavioural testing. Mice had free access to food and water throughout the testing schedule. Before each test, mice were taken from their holding room and allowed to habituate to the testing room for 5–10 min prior to testing. All behavioural experiments were performed during the light phase and all apparatus were cleaned with 70% ethanol before use and between trials. General locomotor activity and anxiety-like behaviour was assessed using the open field test. Mice were individually placed into a Perspex open field arena (30 × 30 × 21 cm) and allowed to freely explore for 10 min. The mice were tracked using a computerised tracking system (ANY-Maze, Version 4.99 m, Stoelting Co., U.S.A.) on a Hewlett-Packard ProBook running Windows 8.1, which automatically recorded distance, average speed, time spent in the two zones (outer perimeter; centre zone) and time spent freezing.

**Object recognition task**. Long-term recognition memory was assessed using the object recognition task. The object recognition task was performed in a rectangular plastic arena (38 × 43 × 18 cm). Three similarly sized objects were used for all mice (plastic culture tube filled with NaCl; brown glass bottle filled with water; 50 ml plastic tube filled with solution of bromophenol blue) with animals showing no preference for one particular object over the others. The task consisted of two 3-minute sessions:

1. Familiarisation session: Two identical objects are positioned at two fixed locations within the testing arena.
2. Test session: One of the objects from the Familiarisation session is placed in the same location as before (Familiar object) and one object is replaced with a second object (Novel object).

At the beginning of each session, the test mouse was placed in the arena facing the wall, equidistant from the two objects. The test started automatically when the mouse was placed into the arena and the animal's movement was recorded by ANY-Maze as above except that two zones were drawn around the objects in the arena. Between sessions, the arena and objects were cleaned with 70 % ethanol and the mouse was returned to its home cage for the intersession interval of 3 h. Each mouse's performance was assessed by calculating their discrimination index during the test session:

$$\text{Discrimination index} = \frac{\text{Time}_{\text{Novel}} - \text{Time}_{\text{Familiar}}}{\text{Time}_{\text{Novel}} + \text{Time}_{\text{Familiar}}}$$

**Y-maze**. Working spatial memory was assessed using the spontaneous alternation on a Y-Maze. Individual mice were placed in the centre zone of a Perspex Y-Maze (three 30 ×5 cm joined by a 20 cm diameter central area; bounded by wall 15 cm high) and allowed to freely explore the apparatus for 8 min. Arm entries were counted by the experimenter when the mouse placed all four paws into an arm, delineated by the slot for guillotine doors associated with the apparatus. Each arm entry was recorded in the order in which they occurred. Spontaneous alternation was assessed after the experiment was over; a successful alternation being defined as the mouse entering all three arms of the Y-Maze over any 4-arm entry span. Errors occurred when a mouse exited and entered the same arm before visiting either of the other two arms.

**Electroencephalography (EEG)**. Mice were anesthetized with isoflurane (isoflurane; 5% induction, 1–2% maintenance) and placed in a mouse-adapted stereotaxic frame. Body temperature was maintained within the normal physiological range with a feedback-controlled heat pad (Harvard Apparatus, Kent, UK; Holliston, MA). After making a midline scalp incision, the bregma was located, and three partial craniectomies were performed for the placement of skull-mounted recording screws (Bilaney Consultants). The electrode assembly was fixed in place

with dental cement, and the mouse was placed in a heated chamber for surgery recovery. After full recovery, mice were placed in an open Perspex box, which allowed free movement. The EEG was recorded using a Xltek® brain monitor amplifier (Natus). Mouse EEG data were analysed and quantified using LabChart 8 software (ADInstruments, Oxford, U.K.). Seizures were defined as high-amplitude (>2× baseline) high-frequency (>5 Hz) polyspike discharges lasting >5 s. Status epilepticus was defined as at least 5 min of continuous seizure activity. From the EEG recordings we calculated the total power and % of total power as previously described[49]. EEG total power was plotted as percentage of baseline recording (each animal's EEG power post seizure compared to its own baseline EEG).

**Cell culture**. The human brain endothelial cell line (hCMEC/d3) was cultured in EGM2-MV media on collagen I coated culture plates. The mouse brain endothelial cell line (b.End3) was cultured in DMEM Glutamax supplemented with 10% foetal bovine serum (FBS).

**MTS assay**. bEnd.3 cells were seeded into 96-well plates at a density of 2 ×10^5 cells per well in DMEM. After 24 h, cells were transfected with 5 pmol non-targeting (siNT) or claudin-5 targeting (siCldn5) siRNA using Lipofectamine 2000 (Invitrogen). At the indicated time points, the medium was replaced and 20 μl of CellTiter 96 AQueous One Solution Cell Proliferation Assay (Promega) was added to each well. The reagent was incubated for 2 h, and the concentration of the formazan product was measured in a spectrophotometer at 450 nm.

**Measurements of monolayer integrity**. Transendothelial electrical resistance (TEER) measurements were recorded using the EndOhm ohmmeter with chopstick electrodes after cells had reached confluence in transwell inserts. After stable TEER measurements, cells were treated with 0.1 μM, 1 μM, 10 μM or 100 μM of the TGFβ inhibitor RepSox for 24 h. Monolayer flux of FITC-dextran (4–70 kDa) was determined by adding 1 mg/ml of FITC-dextran solution to the apical compartment for 1 h. Fluorescence intensity was determined from the basolateral compartment with a Fluostar Optima fluorescent plate reader. Blank corrected relative fluorescent units were normalised to untreated wells.

**Scratch assay**. bEnd.3 cells were seeded into 12-well plates at a density of 2 ×10^5 cells/well. After 24 h, cells were transfected with 20 pmol non-targeting (siNT) or claudin-5 targeting (siCldn5) siRNA. After 24 h, the medium was replaced and the cells were scratched down the middle with a 200 μl pipette tip, washed two times with PBS and wound closure was imaged with a digital camera attached to a brightfield microscope at 0 h, 24 h and 48 h intervals.

**Real-time qPCR**. RNA was isolated by Trizol extraction and cDNA was reverse transcribed from RNA (500 ng) with the High-Capacity cDNA Reverse Transcription Kit (Applied Biosystems). Transcript levels were quantified on the StepOne Plus instrument (Applied Biosystems) with FastStart Universal SYBR Green Master (ROX) master mix (Roche). The RT-PCR reaction conditions were as follows: 95 °C × 2 min, (95 °C × 5 s, 60 °C × 30 s) ×40, 95 °C × 15 s, 60 °C × 1 min, 95 °C × 15 s, 60 °C × 15 s. The primer sequences for the RT-PCR experiments are supplied in Supplementary Table 2. Relative gene expression levels were measured using the comparative $CT$ method ($\Delta\Delta CT$). Expression levels of target genes were normalised to the housekeeping gene β-actin.

**Immunohistochemistry**. Mice were sacrificed by cervical dislocation and brains were quickly frozen in OCT compound (Fisher Scientific). Surgically resected brain tissue from TLE patients was quickly frozen on dry ice, embedded in OCT compound and 20 μm sections from the lateral temporal lobe, amygdala, hippocampus and temporal pole or autopsy control sections were fixed in methanol for 10 min at −20 °C. Sections were rehydrated in PBS and blocked in 5% normal goat serum/ 0.1% Triton X-100 for 30 min at room temperature. Antibody incubations were performed overnight at 4 °C. Secondary fluorescent conjugated antibodies were incubated for 1 h at 37 °C. Nuclei were stained with Hoechst for 1 min and slides were coverslipped with Aqua Poly/mount. Images were acquired on a Zeiss LSM 710 confocal microscope (Carl Zeiss). Primary antibodies used were rabbit anti-claudin-5 (1/500 Life Technologies, #34-1600), rabbit anti-CD31 (1/100 Abcam, #ab28364), rat anti-CD31 (1/100 BD Biosciences, #550274) rabbit anti-IBA1 (1/500 Wako, #019-19741), rat anti-ICAM1 (1/100 Biolegend, #116101), rabbit anti-human fibrinogen FITC (1/100 DAKO, #F0111), goat anti-human IgG Cy3 (1/100 Abcam, #ab97170), mouse anti-GFAP (1/500 Sigma, #G3893), isolectin GS-B4 Alexa Fluor 568 (1/300 Biosciences, #I21412), goat anti-mouse Cy3 (1/500 Biosciences #A10521), goat anti-rabbit IgG Alexa Fluor 488 (1/500 Abcam, #ab150077), goat anti-rat Alexa Fluor 594 (1/500 Abcam, #ab150160).

**Biotin permeability**. EZ-link Sulfo-NHS-SS-Biotin (VWR) was re-suspended to 2 mg/ml in sterile saline and injected by tail vein at a dose of 20 mg/kg for 10 min before mice were sacrificed. Brains were fixed overnight in 4% formaldehyde, cryoprotected in 30% sucrose and flash frozen in OCT. 20 μm sections were

incubated in streptavidin Cy3 overnight at 4 °C, washed 3 times in PBS/0.1 % triton x-100, stained with Hoechst for 1 min and coverslipped with Aqua Poly/mount.

**Injection of drugs**. The small molecule inhibitor of ALK5, RepSox (STEMCELL), was dissolved in dimethyl sulfoxide and sterile filtered. Stock solutions were then diluted in 0.9% NaCl and mice were injected with 10 mg/kg by i.p. injection 1 h before intrahippocampal injection of kainic acid and then every other day for 7 days. Mice were sacrificed 24 h or 7 days later and processed for immunohistochemistry.

**ELISA**. ELISA's for mouse S100β were carried out with serum using the Duoset ELISA (R&D) according to manufacturer's instructions. 96 well plates were coated with capture antibody overnight at room temperature and then washed three times in PBS plus 0.1% Triton X-100 (PBST). Plates were blocked in 1% BSA for 1 h at room temperature. 50 μl of neat serum or standards was incubated for 2 h at room temperature on 96 well plates. Following three washes in PBST 50 μl of detection antibody was incubated for 2 h at room temperature. Following three washes in PBST 50 μl of streptavidin-HRP was incubated for 20 min at room temperature. Following three washes in PBST 50 μl of substrate solution was added to each well and incubated for 20 min at room temperature in the dark. The reaction was then stopped with 50 μl of stop solution and the absorbance was measured on a plate reader at 450 nm. Blank corrected levels of S100β were calculated from a standard curve using 4 parameter logistic regression.

**Image analysis**. Claudin-5-stained vessel length was determined in ImageJ by converting greyscale images to skeleton with the skeletonise function and analyse skeleton function. Vessel length was summed from 3 images from 3 consecutive sections from Bregma −2.4 to −2.1 mM for a total of 9 measurements per mouse. Microglia morphology was determined with the skeletonise function as previously described[50]. Briefly, 8-bit images were converted to greyscale and brightness/contrast settings were adjusted until all microglia processes could be identified. An Unsharp Mask Filter was applied, and noise was removed with the despeckle function. The image was then converted to binary. The despeckle function was applied followed by close and remove outlier functions. The image was skeletonised, and the skeleton was analysed with the AnalyzeSkeleton function. The data was exported to Excel where it was trimmed to exclude artifacts and the number of endpoint voxels and branch length was summed. Branch length and endpoints was divided by the number of microglia soma to calculate branch length and endpoints/cell. The number of cFos positive cells was calculated with the *"analyse particles"* function in ImageJ. Mean GFAP signal was determined in ImajeJ, adjusting the threshold until all GFAP-positive astrocytes could be identified, then converting the image to binary. Mean GFAP signal was then measured in the dentate gyrus, CA1 and CA3 regions of the hippocampus.

**Dynamic contrast-enhanced magnetic resonance imaging (DCE-MRI)**. All ethical approvals were in place prior to initiation of studies on human subjects. BBB permeability maps were created using the slope of contrast agent concentration in each voxel over time, calculated by a linear fit model as previously described[51]. Thresholds of high permeability was defined by the 95th percentile of all slopes in a previously examined control group. All imaging was performed using a 3T Philips Achieva scanner and included a T1- weighted anatomical scan (3D gradient echo, TE/TR = 3/6.7 ms, acquisition matrix 268 × 266, voxel size: 0.83 × 0.83 × 0.9 mm), T2-weighted imaging (TE/TR = 80/3000 ms, voxel size: 0.45 × 0.45 × 0.4 mm), FLAIR (TE/TR = 125/11000 ms, voxel size:0.45 × 0.45 × 4 mm). For the calculation of pre-contrast longitudinal relaxation time (T10), the variable flip angle (VFA) method was used (3D T1w-FFE, TE/TR = 2.78/5.67 ms, acquisition matrix: 240 × 184, voxel size: 0.68 × 0.68 × 5 mm, flip angles: 2, 10, 16 and 24°). DCE sequence was then acquired (Axial, 3D T1w-FFE, TE/TR = 2.78/5.6 ms, acquisition matrix: 240 × 184, voxel size: 0.68 × 0.68 × 5 mm, flip angle: 6°, Tt = 6.5 s, temporal repetitions: 61, total scan length: 7.6 min). An intravenous bolus injection of the contrast agent gadobentate dimeglumine (Gd-BOPTA, Bracco Diagnostics Inc., Milan, Italy) was administered using an automatic injector after the first three DCE repetitions. Case details are summarised in Table 1 in supplementary information.

**Statistical analysis**. Behavioural tests were performed with automated tracking systems when possible. If not (for Racine scoring and y-maze), scoring was done by experimenters blinded to experimental conditions. Outliers for behavioural testing after kainic acid or viral-mediated manipulations were identified as being greater than 2 SD from the mean and excluded from statistical analysis. Normality was determined by D'Agostino–Pearson, Shapiro–Wilk and Kolmogorov–Smirnov normality tests using GraphPad Prism software (version 9.0). Most datasets were normally distributed, and statistical comparisons were made with $t$-tests, one-way ANOVAs, two-way ANOVAs, and Pearson's correlations with GraphPad Prism software (version 9.0). Bonferroni was used as a post hoc test when appropriate for one-way and two-way ANOVAs and statistical significance was set at $p < 0.05$. If datasets were not normally distributed a non-parametric Mann−Whitney or Kruskal−Wallis test was used for two or three groups, respectively. Statistical

significance was set at $p < 0.05$ with $*p < 0.05$; $**p < 0.01$; $***p < 0.001$; $****p < 0.0001$. All quantitative PCR, ELISA, flux assays, TEER measurements and MTS assays were performed in duplicate.

**Reporting summary**. Further information on research design is available in the Nature Research Reporting Summary linked to this article.

## Data availability

All data supporting the findings of this study are available within the paper and Supplementary Information files. Source data are provided with this paper.

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

## Acknowledgements
This work was supported by grants from Science Foundation Ireland (SFI), (12/YI/B2614 and 11/PI/1080), The Irish Research Council (IRC) and by a research grant from SFI under grant number 16/RC/3948 and co-funded under the European Regional Development fund by FutureNeuro industry partners. The Campbell lab is also supported by a European Research Council (ERC) grant, "Retina-Rhythm" (864522). We thank Charles Murray and Ciaran Gavagan for animal husbandry.

## Author contributions
C.G and M.C: designed research. C.G and Ni.H: performed experiments. C.R: EEG analysis. M.A.M and Ch.B: Generation of mouse models. A.R: in vitro experiments. R.C, C.B and C.P.D: Patient recruitment and clinical evaluation. E.O.K: MRI analysis. I.B: in vitro experiments. N.H and C.D: Genotyping. M.A.F, D.F.OB, J.C, F.M.B and A.B: Sample collection and neuropathology. D.H: Tissue collection and supervision. C.G and M.C analysed the data and wrote the manuscript which was edited by all authors.

## Competing interests
Trinity College Dublin owns an intellectual property portfolio related to the regulation of claudin-5 to treat epilepsy. M.C., C.G., N.H. and C.D. are named inventors. All other authors declare no competing interest.
