## [Peer Review File · Nature Communications]

Microvascular stabilization via blood-brain barrier regulation prevents seizure activityREVIEWER COMMENTS

Reviewer #1 (Remarks to the Author):

Drug resistant epilepsy is a major health problem, and treatments to influence the course of epilepsy and novel methods to suppress seizures are needed. It would be useful to have a roadmap summarized as to how this line of research may be pursued in humans, and the possible adverse effects of attempts to upregulate Claudin-5.

An area that should also be discussed is the concept that AED resistance is the result of upregulation of BBB, so that AED cannot access the part of the brain that is generating seizures. How do the authors reconcile that hypothesis with their findings?

Reviewer #2 (Remarks to the Author):

The paper by Green et al addresses the role of BBB disruption, mostly via claudin-5 alterations, in TLE. They made strong effort to obtain a large number of human samples, which is highly appreciated. Although the topic is important and timely, and the authors used a large array of approaches, I have several concerns that reduce my enthusiasm for the study in its present form. In a nutshell, the biggest significance of the work lies in figure 6, but data for this figure needs significant improvement. The other figures describe that claudin-5 is altered in TLE patients and a mouse model of TLE seizures, and decreasing claudin-5 makes BBB alterations and seizure worse. Major concerns:

1. Fig. 1 and Fig S1: There is not a big decrease in claudin 5. In addition, since it is TLE, should we not see more dramatic changes in the hippocampus compared to the cortex? why is the cortex affected? is this the result of seizures?

The claudin-5 western blot does not seem that convincing. 3 samples in autopsy are higher, but the other 3 are similar to the TLE patients. Analysis does not seem easy to perform based on the quality of the band. Could the authors rerun the blots?

2. The authors do not have enough mice for behavior. I should be at least 10!

3. Why did the authors look at permeability at 48hrs post injection of AAV shRNA claudin-5 and not 3h or 24 hrs like the previous experiments in Fig. 2?

4. Fig. 4g and g: "seizure behaviour and astrogliosis as evidenced by GFAP positivity were significantly reversed (Fig 4f & g)." Where is the seizure data?

5. Figure 5: "qPCR analysis revealed that claudin-5 depletion induced expression of Icam1". Based on the data, it does not seem that ICAM1 expression was increased with claudin-5 knockdown.

6. Figure 6: the authors should better describe the human data considering that the molecules are not always upregulated in the hippocampus, which is surprising for TLE cases.

7. Figure 6: It may be worse examining the impact of reintroducing claudin 5 on BBB and seizures. At least it is important to better assess the function of RepSox at a later time point and on seizures. It seems that the authors may have checked on seizures in Fig. S4d, but it is not described in the results.

Minor concerns:

Please mention which type of seizure disorders the humans had in the abstract.

Reviewer #3 (Remarks to the Author):

This manuscript presents some compelling and novel work that furthers our understanding of the relationship between claudin-5 and epilepsy in human patients and mice. However, we have several major concerns: 1) some of the claims of the paper are unsupported by the data, 2)

quantification of data is often inappropriate, inconsistent across figures, or absent, and 3) many methods are incompletely described or entirely absent. Overall, despite a plethora of major concerns outlined below, the authors present strong evidence in Figures 1, 3, and 4 of the mechanistic link between claudin-5 loss and seizures that warrant publication in a more focused manuscript. We do not believe the data from Figures 2, 5, and 6 (with the possible exception of 2C and 2I) should be included in the revised manuscript, for reasons described below.

Major concerns (Figures 1, 3 and 4 & general concerns):

Figure 1

1A-C - Representative control images from healthy patients should be included, as well as annotations of the relevant regions and phenotypes of interest, to clarify the key anatomical and functional differences between healthy patient controls and TLE patients. In addition, the authors should show representative images of the pathologies described in Table S1 and quantify the data.
1E - To assess leakage of endogenous serum proteins, the authors need to use an independent vessel marker to determine whether these proteins are within or outside the confines of the brain vasculature; Claudin-5 (used in Figure 1E) is not an appropriate vessel marker since the authors clearly show in Figure 1D that its levels are reduced in TLE; the same applies for their IgG, fibrinogen, and albumin images in Figure S1. Lastly, the authors should ensure that the vessels they use for analysis present with normal capillary morphology. At present, many of the TLE vessel representative images look like larger vessels than shown in control images.
The authors must include a methods section that details patient and healthy control recruitment, relevant demographics (including age), and any associated approval processes for working with patient data.

Figure 3

3H - A major concern with the kainic acid-induced seizure model is that sub-convulsive dosages of kainic acid are able to induce status epilepticus in control mice, indicating that they are in fact not sub-convulsive. The authors should change the language to "mildly convulsive".
3I - The authors should include a methods section for their observation and quantification of the Racine score. How long were these mice observed? Were they scored manually in real time? Captured on video? Using a single trial or over multiple days?

Figure 4

4E - The authors should segment individual cells and count the number/density of cell bodies to distinguish between changes in cell morphology and cell proliferation. Integrated density alone is insufficient to prove hyperproliferation.

Discussion

The authors provide evidence of claudin-5 downregulation following kainic acid-induced seizures (Figure 2C) and claudin-5 KD preceding spontaneous recurring seizures (Figure 4C), prompting the question of which phenomenon (claudin-5 reduction vs. seizures) is a cause vs. a consequence of the other. The authors should address these intriguing yet perplexing data (e.g. the possibility of a positive feedback loop) in the discussion.

Methods

There are a number of methods in the paper that are poorly explained or not explained at all. Specific details are noted above, but in general, the methods should include complete dosing information for tracer experiments (e.g. not just 2 mg/mL for biotin leakage assays but also the dosage or volume of tracer). They should explain in detail their quantification methods (e.g. GFAP quantification in Figure 4F/G), including their behavioral assays (Figure 2 and Figure S4) and seizure severity (e.g. manual vs. automated scoring, observation period length, videography, etc.) Data, such as claudin-5 and GFAP expression, should be quantified the same way across figures.

Major concerns (Figures 2, 5 and 6):

Figure 2

We do not believe this data warrants inclusion in the final manuscript because the central claim that "focal induction of seizures induces claudin-5 disruption at the BBB" is not supported by the

data. The authors could consider including the data showing loss of claudin-5 and BBB leakage (2C, 2I) after focal kainic acid if they properly quantify the data and show control images.

2A - The n's should be increased for the chronic treatment condition to enable comparison to the saline and acute conditions. Additionally, the behavioral data could be due to a change in locomotor activity rather than anxiety behavior, so the authors must control for total locomotion in their behavioral analysis.

2B - The authors should substantiate the claim that there is vascular remodeling present in the acute and chronic conditions by further describing and quantifying this phenotype (e.g. branch points or vessel density in the hippocampal sub-regions [CA1, CA3, dentate gyrus] vs. surrounding regions). Furthermore, given the evidence of decreased claudin-5 levels in human TLE (Figure 1D,F) and kainic acid-evoked seizures (Figure 2C-D), the authors should use an independent vessel marker other than claudin-5 for these analyses.

2C - Control images of claudin-5 staining should be included.

2D - To enable comparison with the human TLE patient data in Figure 2F, claudin-5 levels in the chronic and acute conditions should also be quantified as area fraction of CD31 or another vessel marker.

2E-G - The n's for the acute and chronic conditions in the qPCR experiments should be increased to n=6 to match the PBS-injected controls.

2I - Leakage in this figure should be quantified, ideally as area of leakage outside vessel area.

2H - The authors should cite a source that definitively shows that serum S100b is a direct result of brain-blood leakage through the BBB.

2K - Claudin-5 protein levels should be quantified as in Figure 1F or Figure 2D to maintain consistency in quantification methods. In addition, a clear junctional staining pattern should be demonstrated.

2N - The data could be a result of decreased locomotion in knockdown mice rather than an anxiety phenotype, as the authors conclude. In addition to the distance traveled in open field, the authors should also show and quantify the time in the center zone to enable comparison with the model presented in Figure 2A

2O - The authors should functionally validate their claudin-5 KD model using an additional metric outside of sodium fluorescein fluorescence in homogenized tissue since it is highly concerning that saline-injected AAV-shRNA-Cldn5 mice do not display an increase in sodium fluorescein leakage in comparison to saline-injected AAV-shRNA controls, given the authors' other findings of small tracer leakage under conditions of claudin-5 reduction. We suggest assessing biotin leakage with histology like in Figure 2I, with quantification.

2P - Representative staining (with quantification) should also be shown for GFAP for the sake of consistency and coherence across the paper and to enable comparison with the models/data in Figure S2 and Figure 3O

In the methods section, the authors must specify the AAV serotype they use for claudin-5 KD, along with the source of the virus and the shRNA sequences they use. If made in-house, the authors should include the prep and titering protocol. Without these details, readers have no way of knowing which cell type(s) the virus targets.

Figure 5

We do not think these data add any mechanistic insight to the paper, so they should not be included. We disagree with the interpretation of cell motility as evidence that claudin-5 has roles outside tight junctions, simply because the presence of tight junctions between cells prevents motility -- so loss of TJs could reasonably coincide with increased motility.

The existing data could be strengthened by adding immunohistochemistry data for the leukocyte adhesion molecules and staining for leukocytes in the brain parenchyma to substantiate the claim that there is "immune cell infiltration into the brain parenchyma".

The interaction between IL1B and claudin-5 is not supported by the data. The authors show an additive effect of claudin-5 knockdown and IL1B on transcriptional changes but the data do not indicate any specific relationship between the two molecules.

If the authors are interested in ICAM1 expression changes, they could provide more convincing data by showing upregulation of ICAM1 staining in mouse brain after Cldn5 KD.

Figure 6

We do not think this figure supports the central claim that "Targeted elevation of claudin-5 expression stabilizes the BBB and decreases neural damage in kainic acid induced epilepsy"

because their experimental manipulation (RepSox drug delivery) is not a targeted elevation of claudin-5. Alk5 is present in non-endothelial cell types (e.g. He et al. (2014) doi:10.1038/nn.3732) so the current experiments are insufficient to draw a mechanistic link between loss of claudin-5 in BBB and a protective effect of RepSox on seizure activity.

The most important functional assay to test the therapeutic effects of RepSox is intrahippocampal kainic acid-induced seizure activity, which the authors fail to show. To demonstrate a direct functional link, the authors would need to show that RepSox is protective against kainic acid-induced seizures in claudin-5 KD mice.

6G - The authors should address the glaring discrepancy between the fact that RepSox does not induce expression of Ocln and Tjp2 mRNA (Figure S4) but is somehow able to rescue TGF β -induced loss of Ocln and Tjp2 mRNA.

6I - Images of IgG and NaFl permeability should be counterstained with a vessel marker (e.g. CD31) to clearly demonstrate tracer extravasation. The authors should also employ higher magnification images as in Figure 2I to clearly show whether tracers are contained within the brain vasculature or leak out.

6J - cFOS staining should be quantified as a % of hippocampal neurons to account for the regional/morphological variability in neuronal density across images/slices.

Figure S3

S3D: The blot looks incompletely transferred, artificially lowering the ICAM-1 intensity in WT mice (except for the WT + KA directly next to Cldn5 deficient samples). It must be repeated to make any conclusion.

We thank each reviewer for their careful critique of our paper. We believe their observations and comments have added to and improved this study. Detailed below in “bold” are responses to each individual comment.

Reviewer #1 (Remarks to the Author):

Drug resistant epilepsy is a major health problem, and treatments to influence the course of epilepsy and novel methods to suppress seizures are needed.

It would be useful to have a roadmap summarized as to how this line of research may be pursued in humans, and the possible adverse effects of attempts to upregulate Claudin-5.

Claudin-5 has been found to be dysfunctional in many neurological disorders, but the effects of claudin-5 upregulation have not been studied. Based on our findings, the next logical step will be to assess the therapeutic efficacy of modulating claudin-5 in epilepsy models via small molecule regulation or gene therapy delivery of claudin-5 cDNA. If this proves beneficial in rodent models of epilepsy then the therapy could be trialed in non-human primates prior to clinical deployment.

An area that should also be discussed is the concept that AED resistance is the result of upregulation of BBB, so that AED cannot access the part of the brain that is generating seizures. How do the authors reconcile that hypothesis with their findings?

We have now discussed the potential of upregulation of BBB properties as a contributing factor to the process of epileptogenesis in the discussion section with particular emphasis on the role of drug metabolising enzymes and efflux transporters.

Reviewer #2 (Remarks to the Author):

The paper by Green et al addresses the role of BBB disruption, mostly via claudin-5 alterations, in TLE. They made strong effort to obtain a large number of human samples, which is highly appreciated. Although the topic is important and timely, and the authors used a large array of approaches, I have several concerns that reduce my enthusiasm for the study in its present form. In a nutshell, the biggest significance of the work lies in figure 6, but data for this figure needs significant improvement. The other figures describe that claudin-5 is altered in TLE patients and a mouse model of TLE seizures, and decreasing claudin-5 makes BBB alterations and seizure worse.

Major concerns:

1. Fig. 1 and Fig S1: There is not a big decrease in claudin 5. In addition, since it is TLE, should we not see more dramatic changes in the hippocampus compared to the cortex?

The surrounding cortex is affected due to the spreading nature of the seizure. This is evidenced in Figure 1a, where we show cortical gadolinium extravasation in the patients with TLE patients and in Fig S1b where clear temporal lobe atrophy is observed. In the context of neuroinflammation and gliosis, we do see more dramatic changes in the hippocampus compared to the cortex and this data is now included in (Fig S1b).

Why is the cortex affected? is this the result of seizures?

Yes, as mentioned above, spreading depolarisation in surrounding cortical regions likely results in vascular disruption to the adjacent cortex.

The claudin-5 western blot does not seem that convincing. 3 samples in autopsy are higher, but the other 3 are similar to the TLE patients. Analysis does not seem easy to perform based on the quality of the band. Could the authors rerun the blots?

We have repeated the Western blot for claudin-5 with 7 samples from TLE patients and the new blot and quantification can be found in Fig S1e.

2. The authors do not have enough mice for behavior. I should be at least 10!

Behavioural experiments have now been repeated to increase n number to at least 10.

3. Why did the authors look at permeability at 48hrs post injection of AAV shRNA claudin-5 and not 3h or 24 hrs like the previous experiments in Fig. 2?

The rationale for looking at 48 hours was due to the different experimental design. In Fig 2 a-l we used intrahippocampal injection of kainic acid while figure 2 j-p used intraperitoneal injection. The different routes of administration lead to quite different dynamics of BBB permeability. As shown in Munji *et al* PMID 31611708, i.p. kainic acid leads to a peak in BBB permeability at 48 hours, which was our reason for choosing this timepoint.

4. Fig. 4g and g: "seizure behaviour and astrogliosis as evidenced by GFAP positivity were significantly reversed (Fig 4f & g)." Where is the seizure data?

Seizure data has now been included in Fig 4f

5. Figure 5: "qPCR analysis revealed that claudin-5 depletion induced expression of Icam1". Based on the data, it does not seem that Icam1 expression was increased with claudin-5 knockdown.

We now show in Fig 5 i-j that loss of claudin-5 leads to a dose dependent increase in Icam-1 which correlates with expression of claudin-5 *in vitro*. Furthermore, in Fig 5 k-l, we show that claudin-5 knockout mice subjected to kainic acid-induced seizures as in Fig 3 show a dramatic upregulation in Icam-1 and leukocyte extravasation in the brain (Fig 5k-l).

6. Figure 6: the authors should better describe the human data considering that the molecules are not always upregulated in the hippocampus, which is surprising for TLE cases.

In Figure 6 we do not show any human hippocampus data, however in the revised supplementary Figure 6 we show that TGFB1 is 5-fold upregulated in the hippocampus along with significant increases in Snail1 and Smad2 transcripts, hence our reasoning for interrogating TGFB signalling in Figure 6.

7. Figure 6: It may be worse examining the impact of reintroducing claudin 5 on BBB and seizures. At least it is important to better assess the function of RepSox at a later time point and on seizures. It seems that the authors may have checked on seizures in Fig. S4d, but it is not described in the results.

We agree that examining the impact of claudin-5 overexpression on BBB and seizures is the next logical step that warrants future studies. We have included additional data on the effects of RepSox on the brain following kainic acid injection. In Fig 6i we show that RepSox attenuates

seizures as measured by the Racine scale. Furthermore, at chronic timepoints, RepSox attenuates astrogliosis and IgG permeability as shown in Fig S6d-e. We have expanded on these findings in the results section.

Minor concerns:

Please mention which type of seizure disorders the humans had in the abstract.

Seizure types have been included in the demographics table supplementary table 1.

Reviewer #3 (Remarks to the Author):

This manuscript presents some compelling and novel work that furthers our understanding of the relationship between claudin-5 and epilepsy in human patients and mice. However, we have several major concerns: 1) some of the claims of the paper are unsupported by the data, 2) quantification of data is often inappropriate, inconsistent across figures, or absent, and 3) many methods are incompletely described or entirely absent. Overall, despite a plethora of major concerns outlined below, the authors present strong evidence in Figures 1, 3, and 4 of the mechanistic link between claudin-5 loss and seizures that warrant publication in a more focused manuscript. We do not believe the data from Figures 2, 5, and 6 (with the possible exception of 2C and 2I) should be included in the revised manuscript, for reasons described below.

Major concerns (Figures 1, 3 and 4 & general concerns):

Figure 1

1A-C - Representative control images from healthy patients should be included, as well as annotations of the relevant regions and phenotypes of interest, to clarify the key anatomical and functional differences between healthy patient controls and TLE patients. In addition, the authors should show representative images of the pathologies described in Table S1 and quantify the data.

Control images from a healthy (non-diseased) participant have been included in Fig S1a. Phenotypes associated with TLE including temporal lobe atrophy and reduced hippocampal volume have also been included in Fig S1b. Unfortunately, we have been unable to receive the slides from which the pathologies were diagnosed as this was performed in the hospital following surgical resection and was not released for clinical research.

1E - To assess leakage of endogenous serum proteins, the authors need to use an independent vessel marker to determine whether these proteins are within or outside the confines of the brain vasculature; Claudin-5 (used in Figure 1E) is not an appropriate vessel marker since the authors clearly show in Figure 1D that its levels are reduced in TLE; the same applies for their IgG, fibrinogen, and albumin images in Figure S1. Lastly, the authors should ensure that the vessels they use for analysis present with normal capillary morphology. At present, many of the TLE vessel representative images look like larger vessels than shown in control images.

Collagen IV has now been included as an independent vessel marker to clearly show extravascular leakage of endogenous serum proteins. More representative images of capillaries have now been included in Fig 1 and Fig S1d.

The authors must include a methods section that details patient and healthy control recruitment, relevant demographics (including age), and any associated approval processes for working with patient data.

Methods section now includes details on patient recruitments and approval processes for working with patient samples/data. Demographics are included in Table 1.

Figure 3

3H - A major concern with the kainic acid-induced seizure model is that sub-convulsive dosages of kainic acid are able to induce status epilepticus in control mice, indicating that they are in fact not sub-convulsive. The authors should change the language to "mildly convulsive".

As suggested, we have changed the language to "mildly convulsive".

3I - The authors should include a methods section for their observation and quantification of the Racine score. How long were these mice observed? Were they scored manually in real time? Captured on video? Using a single trial or over multiple days?

Methods now include details on scoring seizure behaviour. In short, mice were observed for 2 hours following kainic acid injection during which time they were captured on video in addition to being manually scored by an observer blind to treatment. A second individual blind to treatment then scored the video data.

Figure 4

4E - The authors should segment individual cells and count the number/density of cell bodies to distinguish between changes in cell morphology and cell proliferation. Integrated density alone is insufficient to prove hyperproliferation.

This data is now included in Fig 4E to replace integrated density measurements.

Discussion

The authors provide evidence of claudin-5 downregulation following kainic acid-induced seizures (Figure 2C) and claudin-5 KD preceding spontaneous recurring seizures (Figure 4C), prompting the question of which phenomenon (claudin-5 reduction vs. seizures) is a cause vs. a consequence of the other. The authors should address these intriguing yet perplexing data (e.g. the possibility of a positive feedback loop) in the discussion.

We have expanded on these interesting points in the discussion section.

Methods

There are a number of methods in the paper that are poorly explained or not explained at all. Specific details are noted above, but in general, the methods should include complete dosing information for tracer experiments (e.g. not just 2 mg/mL for biotin leakage assays but also the dosage or volume of tracer). They should explain in detail their quantification methods (e.g. GFAP quantification in Figure 4F/G), including their behavioral assays (Figure 2 and Figure S4) and seizure severity (e.g. manual vs. automated scoring, observation period length, videography, etc.)

Data, such as claudin-5 and GFAP expression, should be quantified the same way across figures.

Accurate dosing information has now been included for tracer experiments along with quantification methods and behavioural protocols in the methods section. IHC for claudin-5 and GFAP has now been quantified the same for each figure.

Major concerns (Figures 2, 5 and 6):

Figure 2

We do not believe this data warrants inclusion in the final manuscript because the central claim that “focal induction of seizures induces claudin-5 disruption at the BBB” is not supported by the data. The authors could consider including the data showing loss of claudin-5 and BBB leakage (2C, 2I) after focal kainic acid if they properly quantify the data and show control images.

2A - The n's should be increased for the chronic treatment condition to enable comparison to the saline and acute conditions. Additionally, the behavioral data could be due to a change in locomotor activity rather than anxiety behavior, so the authors must control for total locomotion in their behavioral analysis.

N numbers have been increased and centre zone entries/time in centre zone and locomotion data is now included in Fig 2a and Fig S2a.

2B - The authors should substantiate the claim that there is vascular remodeling present in the acute and chronic conditions by further describing and quantifying this phenotype (e.g. branch points or vessel density in the hippocampal sub-regions [CA1, CA3, dentate gyrus] vs. surrounding regions). Furthermore, given the evidence of decreased claudin-5 levels in human TLE (Figure 1D,F) and kainic acid-evoked seizures (Figure 2C-D), the authors should use an independent vessel marker other than claudin-5 for these analyses.

We have now included data on vascular remodelling in Fig S3 where we used collagen IV to assess vascular area coverage, number of endpoints, number of junctions and vessel length. We show that there is reduced vessel coverage in CA3 and dentate gyrus regions of the hippocampus in the chronic epileptic state while there is increased vascularisation in surrounding cortical regions.

2C - Control images of claudin-5 staining should be included.

Control images are now included.

2D - To enable comparison with the human TLE patient data in Figure 2F, claudin-5 levels in the chronic and acute conditions should also be quantified as area fraction of CD31 or another vessel marker.

Claudin-5 levels have been quantified as area fraction of CD31.

2E-G - The n's for the acute and chronic conditions in the qPCR experiments should be increased to n=6 to match the PBS-injected controls.

N numbers have been increased to 6 for all groups.

2I - Leakage in this figure should be quantified, ideally as area of leakage outside vessel area.

Quantification of leakage data is now included in Fig 2i.

2H - The authors should cite a source that definitively shows that serum S100b is a direct result of brain-blood leakage through the BBB.

Appropriate citations have now been included.

2K - Claudin-5 protein levels should be quantified as in Figure 1F or Figure 2D to maintain consistency in quantification methods. In addition, a clear junctional staining pattern should be demonstrated.

Claudin-5 protein levels have now been quantified as area fraction of collagen IV and images have been replaced with ones showing clear junctional staining pattern in AAV-shRNA injected mice and disrupted junctional staining in AAV-shRNA-Cldn5 injected mice.

2N - The data could be a result of decreased locomotion in knockdown mice rather than an anxiety phenotype, as the authors conclude. In addition to the distance traveled in open field, the authors should also show and quantify the time in the center zone to enable comparison with the model presented in Figure 2A

N numbers have been increased for this experiment as well as including centre zone entry measurements which shows a clear reduction in AAV-shRNA-Cldn5 injected mice.

2O - The authors should functionally validate their claudin-5 KD model using an additional metric outside of sodium fluorescein fluorescence in homogenized tissue since it is highly concerning that saline-injected AAV-shRNA-Cldn5 mice do not display an increase in sodium fluorescein leakage in comparison to saline-injected AAV-shRNA controls, given the authors' other findings of small tracer leakage under conditions of claudin-5 reduction. We suggest assessing biotin leakage with histology like in Figure 2I, with quantification.

We have now included biotin leakage quantification and images in Fig S4 showing significantly increased extravasation of biotin in AAV-shRNA-Cldn5 injected mice.

2P - Representative staining (with quantification) should also be shown for GFAP for the sake of consistency and coherence across the paper and to enable comparison with the models/data in Figure S2 and Figure 3O

Representative GFAP staining and quantification has now been included in Fig 2P.

In the methods section, the authors must specify the AAV serotype they use for claudin-5 KD, along with the source of the virus and the shRNA sequences they use. If made in-house, the authors should include the prep and titrating protocol. Without these details, readers have no way of knowing which cell type(s) the virus targets.

AAV serotype (AAV2/9) has now been included along with citation to paper listing virus and shRNA sequences.

Figure 5

We do not think these data add any mechanistic insight to the paper, so they should not be included. We disagree with the interpretation of cell motility as evidence that claudin-5 has roles outside tight junctions, simply because the presence of tight junctions between cells prevents motility -- so loss of TJs could reasonably coincide with increased motility.

While loss of tight junctions will influence cellular motility, we believe the central tight junction component involved in this process is claudin-5 as no effect is observed following loss of occludin or Lsr (data not shown).

The existing data could be strengthened by adding immunohistochemistry data for the leukocyte adhesion molecules and staining for leukocytes in the brain parenchyma to substantiate the claim that there is “immune cell infiltration into the brain parenchyma”.

We now include additional experiments showing a dose-dependent increase in Icam-1 mRNA following knockdown of claudin-5 in mouse brain endothelial cells in Fig 5 i-j. Additionally, we show increased vascular and leukocyte Icam-1 expression in claudin-5 het knockout mice in Fig 5 k-l.

The interaction between IL1B and claudin-5 is not supported by the data. The authors show an additive effect of claudin-5 knockdown and IL1B on transcriptional changes but the data do not indicate any specific relationship between the two molecules.

Our rationale for including these experiments was to show that loss of claudin-5 can exacerbate inflammation in endothelial cells. We have shown in Fig 2 and 5 that there is up-regulation of markers associated with gliosis and inflammation following loss of claudin-5. Given the upregulation of inflammatory markers in TLE samples (now included in Fig S1c), we wanted to understand how loss of claudin-5 could influence the expression of such markers.

If the authors are interested in ICAM1 expression changes, they could provide more convincing data by showing upregulation of ICAM1 staining in mouse brain after Cldn5 KD.

We have included this data in the amended Fig 5 as mentioned above.

Figure 6

We do not think this figure supports the central claim that “Targeted elevation of claudin-5 expression stabilizes the BBB and decreases neural damage in kainic acid induced epilepsy” because their experimental manipulation (RepSox drug delivery) is not a targeted elevation of claudin-5. Alk5 is present in non-endothelial cell types (e.g. He et al. (2014) doi:10.1038/nn.3732) so the current experiments are insufficient to draw a mechanistic link between loss of claudin-5 in BBB and a protective effect of RepSox on seizure activity.

The most important functional assay to test the therapeutic effects of RepSox is intrahippocampal kainic acid-induced seizure activity, which the authors fail to show. To demonstrate a direct functional link, the authors would need to show that RepSox is protective against kainic acid-induced seizures in claudin-5 KD mice.

We have performed intrahippocampal kainic acid-induced seizures in mice treated with RepSox and show in Fig 6i that RepSox attenuates kainic acid-induced seizures acutely and prevents chronic astrogliosis and BBB leakage (Fig S6).

6G - The authors should address the glaring discrepancy between the fact that RepSox does not induce expression of Ocln and Tjp2 mRNA (Figure S4) but is somehow able to rescue TGFβ-induced loss of Ocln and Tjp2 mRNA.

RepSox attenuates the activation of TGFβ-signalling resulting in a rescue of the diminished expression of Ocln and Tjp2 independent of RepSox being able to induce expression of either. So,

in the context of cell stress, it is protective of Ocnc and Tjp2 expression, likely as a result of increasing claudin-5 and TJ stabilisation as a whole.

6I - Images of IgG and NaFl permeability should be counterstained with a vessel marker (e.g. CD31) to clearly demonstrate tracer extravasation. The authors should also employ higher magnification images as in Figure 2I to clearly show whether tracers are contained within the brain vasculature or leak out.

Higher magnification images have now been included counterstained with Collage IV to show extravascular leakage of tracers.

6J - cFOS staining should be quantified as a % of hippocampal neurons to account for the regional/morphological variability in neuronal density across images/slices.

cFos staining has now been expressed as % of hippocampal neurons.

Figure S3

S3D: The blot looks incompletely transferred, artificially lowering the ICAM-1 intensity in WT mice (except for the WT + KA directly next to Cldn5 deficient samples). It must be repeated to make any conclusion.

The Western blot has been repeated and is now included in Fig S5.

REVIEWER COMMENTS

Reviewer #1 (Remarks to the Author):

1. The authors' response in the rebuttal to the need for a development roadmap "Claudin-5 has been found to be dysfunctional in many neurological disorders, but the effects of claudin-5 upregulation have not been studied. Based on our findings, the next logical step will be to assess the therapeutic efficacy of modulating claudin-5 in epilepsy models via small molecule regulation or gene therapy delivery of claudin-5 cDNA. If this proves beneficial in rodent models of epilepsy then the therapy could be trialled in non-human primates prior to clinical deployment."

This needs to be clarified in the Manuscript discussion.

2. The response to my initial comment "An area that should also be discussed is the concept that AED resistance is the result of upregulation of BBB, so that AED cannot access the part of the brain that is generating seizures. How do the authors reconcile that hypothesis with their findings?"

Authors Response: In this regard, it has been suggested that localised BBB permeability may contribute to epilepsy which is refractory to medication due to reduced bioavailability of anti-seizure drugs [28]

This does not address in the manuscript the issue that it has been posited that upregulation of BBB, rather than increased permeability, may be a factor in AED resistance.

Reviewer #2 (Remarks to the Author):

The authors did a significant amount of work to address the reviewers' comments.

I still have a few comments:

1. The analysis of the western blot data in Fig. S1e looks odd. Two data points in the TLE condition should be extremely low and one should actually be 0. In the control, one point should be roughly half the mean. Something is fundamentally wrong with analysis.

2. As suggested by Reviewer 3, data included in figure 5 should not be included. It is far from providing a convincing mechanism.

3. Fig 3, Repsox data on seizures are critical. It is unfortunate that for such an important experiment, N=3. Could you add more mice?

As the paper stands, there is no connection between restoring Claudin 5 level/BBB and reduction in seizure activity. Damages to the BBB are expected to worsen seizures and damaging the BBB by removing claudin 5 is expected to induce some form of hyperexcitability. The repsox data are the only one showing a potential rescue.

Reviewer #3 (Remarks to the Author):

The authors addressed most of our concerns, especially the major ones related to methods reporting and quantification. I don't have any more concerns.

We thank each reviewer for their careful perusal of our manuscript. Our responses to each comment are in bold below.

Reviewer #1 (Remarks to the Author):

1. The authors' response in the rebuttal to the need for a development roadmap "Claudin-5 has been found to be dysfunctional in many neurological disorders, but the effects of claudin-5 upregulation have not been studied. Based on our findings, the next logical step will be to assess the therapeutic efficacy of modulating claudin-5 in epilepsy models via small molecule regulation or gene therapy delivery of claudin-5 cDNA. If this proves beneficial in rodent models of epilepsy then the therapy could be trialled in non-human primates prior to clinical deployment."

This needs to be clarified in the Manuscript discussion.

We have now included this as the final discussion point in the manuscript.

2. The response to my initial comment "An area that should also be discussed is the concept that AED resistance is the result of upregulation of BBB, so that AED cannot access the part of the brain that is generating seizures. How do the authors reconcile that hypothesis with their findings?"

Authors Response: In this regard, it has been suggested that localised BBB permeability may contribute to epilepsy which is refractory to medication due to reduced bioavailability of anti-seizure drugs [28]

This does not address in the manuscript the issue that it has been posited that upregulation of BBB, rather than increased permeability, may be a factor in AED resistance.

We have expanded on this topic in the discussion section and would refer the reviewer to Line 289 in the discussion section "While many studies have shown that epilepsy is associated with a breakdown of the BBB, it is also important to recognise that enhancement of BBB properties may also be involved in the development of pharmacoresistance in epilepsy. Drugs designed to act on the brain have been made lipophilic to readily diffuse across the plasma membrane of endothelial cells in the brain, however this makes them potential substrates for efflux carriers at the BBB, especially P-glycoprotein (Pgp). Many studies have indicated that in conditions with a disrupted BBB, including epilepsy, defence mechanisms in brain capillary endothelial cells and perivascular glia may be upregulated, including an increased expression and functionality of Pgp and other efflux carriers and drug-metabolising enzymes [34]. In resected brain tissue from pharmacoresistant epilepsy patients, an elevated level of drug efflux transporters including Pgp and BCRP was observed. Additionally, elevated levels and activity of cytochrome P450 enzymes was found in epileptic brain tissue [11]. Animal studies have revealed that seizures induce expression of Pgp while rats with spontaneous seizures not responding to anti-seizure drugs (ASD) show higher expression of Pgp at the BBB compared to rats responsive to ASDs. Furthermore, animals treated with the Pgp inhibitor tariquidar can be converted to ASD responders

[35-38]. This hypothesis of BBB-associated drug resistance in epilepsy has also been investigated clinically using positron emission tomography (PET) with the PET ligand and Pgp substrate verapamil with and without the Pgp inhibitor tariquidar. In 14 pharmacoresistant TLE patients, 8 seizure free patients and 13 healthy controls, higher Pgp functionality was found in the pharmacoresistant patients compared to seizure-free patients in several brain regions. Higher Pgp activity was associated with higher seizure frequency [39]. Together, this has been suggested to contribute to ASD resistance in epilepsy which affects about 30 % of all patients.”

Reviewer #2 (Remarks to the Author):

The authors did a significant amount of work to address the reviewers' comments.

I still have a few comments:

1. The analysis of the western blot data in Fig. S1e looks odd. Two data points in the TLE condition should be extremely low and one should actually be 0. In the control, one point should be roughly half the mean. Something is fundamentally wrong with analysis.

We thank the reviewer for this observation as there was an error in correcting for background intensity which has now been amended and the figure has been replaced by the correct version.

2. As suggested by Reviewer 3, data included in figure 5 should not be included. It is far from providing a convincing mechanism.

Figure 5f has been removed and the remainder moved to Supplementary Information as suggested by the editor.

3. Fig 3, Repsox data on seizures are critical. It is unfortunate that for such an important experiment, N=3. Could you add more mice?

We agree and have now increased mouse numbers to n = 8 per group showing a significant protection of RepSox against kainic acid induced seizures (P < 0.01).**

As the paper stands, there is no connection between restoring Claudin 5 level/BBB and reduction in seizure activity. Damages to the BBB are expected to worsen seizures and damaging the BBB by removing claudin 5 is expected to induce some form of hyperexcitability. The repsox data are the only one showing a potential rescue.

We do believe that we have shown that restoring claudin-5 levels can prevent seizure activity. In Figure 4, we generated a mouse model that allows us to inducibly express shRNA targeting claudin-5 transcripts. In this regard, we went on to show in Figure 4f and g that attenuation of the expression of this claudin-5 shRNA rescues the hyperexcitability and astrogliosis that was induced by decreased claudin-5 protein. This figure informs us that restoring claudin-5 expression in this inducible knockdown model can rescue seizure activity and the deleterious effects of seizures on the brain. We now also show in Supplementary Figure 7f that RepSox can also rescue kainic acid-induced claudin-5 disruption. We agree with the reviewer that a strategy should be

devised to specifically regulate claudin-5 levels in experimental models of epilepsy (for example AAV-mediated overexpression) to determine its anti-epileptic effects and this will be a fundamental component of future studies. We have therefore expanded on this in the discussion section “Claudin-5 has been found to be dysfunctional in many neurological disorders, but the effects of claudin-5 upregulation have not been studied. Based on our findings, the next logical step will be to assess the therapeutic efficacy of modulating claudin-5 in epilepsy models via small molecule regulation or gene therapy delivery of claudin-5 cDNA. If this proves beneficial in rodent models of epilepsy then the therapy could be trialled in non-human primates prior to clinical deployment”.

Reviewer #3 (Remarks to the Author):

The authors addressed most of our concerns, especially the major ones related to methods reporting and quantification. I don't have any more concerns.

We thank the reviewer for their input to this manuscript.

REVIEWERS' COMMENTS

Reviewer #1 (Remarks to the Author):

My issues have been addressed.

John Duncan

Reviewer #2 (Remarks to the Author):

I am fine with the reviewers' answers.

Response to referees

We are pleased to submit our revised manuscript titled “Microvascular stabilization via blood-brain barrier regulation prevents seizure activity” for consideration as an article in *Nature Communications*.

We would like to thank the three reviewers for their careful assessment of our manuscript as the manuscript is now stronger thanks to their thorough reviews.

Reviewer #1 (Remarks to the Author):

My issues have been addressed.

John Duncan

Response: Thank you Dr Duncan

Reviewer #2 (Remarks to the Author):

I am fine with the reviewers' answers.

Response: Thank you Reviewer 2